

**Time dependent, non-monotonic response of warm convective cloud fields to**
**changes in aerosol loading**
**Guy Dagan, Ilan Koren\*, Orit Altaratz and Reuven H. Heiblum**
Department of Earth and Planetary Sciences, The Weizmann Institute of Science,
Rehovot 76100, Israel.
*\* Correspondence to:* ilan.koren@weizmann.ac.il
**Abstract**
Large Eddy Simulations (LES) with bin microphysics are used here to study cloud
fields' sensitivity to changes in aerosol properties and the time evolution of this
response. Similarly to the known response of a single cloud, we show that the mean
field properties change in a non-monotonic trend, with an optimum aerosol
concentration for which the field reaches its maximal water mass or rain yield. This
trend is a result of competition between processes that encourage cloud development
versus those that suppress it. However, another layer of complexity is added when
considering clouds' impact on the field's thermodynamic properties and how this is
dependent on aerosol loading. Under polluted conditions rain is suppressed, and the
non-precipitating clouds act to increase atmospheric instability. This results in
warming of the lower part of the cloudy layer (in which there is net condensation) and
cooling of the upper part (net evaporation). Evaporation at the upper part of the
cloudy layer in the polluted simulations raises humidity at these levels and thus
amplifies the development of the next generation of clouds (preconditioning effect).
On the other hand, under clean conditions, the precipitating clouds drive net warming
of the cloudy layer and net cooling of the sub-cloud layer due to rain evaporation.
These two effects act to stabilize the atmospheric boundary layer with time
(consumption of the instability). Evolution of the field's thermodynamic properties
affects the cloud properties in return, as shown by migration of the optimal aerosol
concentration toward higher values.



## 1. Introduction

Despite the extensive research conducted in the last few decades, and the fact that
clouds have an important role in the Earth's energy balance (Trenberth et al., 2009)
clouds are still considered to be one of the largest source of uncertainty in the study of
climate and climate change (Forster et al., 2007; Boucher et al., 2013).

Warm cloud (containing liquid water only) formation depends on the availability of
aerosols acting as cloud condensation nuclei (CCN). Changes in aerosol concentration
modulate the cloud droplet size distribution and total number. Polluted clouds
(forming under high aerosol loading) initially have smaller and more numerous
droplets, with narrower size distribution compared to clean clouds (Squires, 1958;
Squires and Twomey, 1960; Warner and Twomey, 1967; Fitzgerald and Spyers-
Duran, 1973).
The initial droplet size distribution affects key cloud processes such as condensation-
evaporation, collision-coalescence and sedimentation. The condensation-evaporation
process is proportional to the total droplet surface area which increases with the
droplet number concentration (for a given total liquid water mass). Under given
supersaturation conditions, the condensation in polluted clouds is more efficient
(Pinsky et al., 2013; Seiki and Nakajima, 2014; Koren et al., 2014; Kogan and Martin,
1994; Dagan et al., 2015a). However, under sub-saturation conditions, due to the same
reason, it implies higher evaporation efficiency. The evaporation induces downdrafts
and stronger vorticity and hence can lead to stronger mixing of the cloud with its
environment in polluted conditions (Xue and Feingold, 2006; Jiang et al., 2006; Small
et al., 2009).
The initiation of collision-coalescence is delayed in polluted clouds (Gunn and
Phillips, 1957; Squires, 1958; Albrecht, 1989). This drives a delay in rain formation
and can affect the amount of surface rain (Rosenfeld, 1999, 2000; Koren et al., 2012;
Khain, 2009; Levin and Cotton, 2009; Dagan et al., 2015b).
Aerosol effects on single warm convective clouds were shown to have an optimal
value with respect to maximal water mass, cloud depth and rain yield (Dagan et al.,
2015a,b). For aerosol concentrations lower than the optimum, the positive relationship
between aerosol concentration and cloud development is a result of two main





processes: 1) larger latent heat release driven by the increase in the condensation
efficiency causing stronger updrafts, and 2) decrease in the effective terminal velocity
($\eta$, i.e. mass weighted terminal velocity of the hydrometeors), (Koren et al., 2015) due
to initial smaller droplets and the delay in the collision-coalescence process. The
smaller droplets have higher mobility (the water mass moves up better with
surrounding updraft), reaching higher in the atmosphere and prolonging the cloud
growth.
For aerosol concentration values above the optimum, the suppressing aerosol effects
take over, namely: 1) stronger mixing of the cloud with its environment driven by the
increased evaporation efficiency (Small et al., 2009), and 2) increased water loading
effect due to the rain suppression.
Understanding of the overall aerosol effect is even more complex when considering
processes on the cloud field scale. Clouds affect the surrounding thermodynamic
conditions by changing the humidity and temperature profiles (Lee et al., 2014;
Seifert et al., 2015; Stevens and Feingold, 2009; Saleeby et al., 2015). In addition,
clouds affect the solar and longwave radiation budgets in the field. Over land the
radiation effects change the surface temperature and therefore can significantly affect
heat and moist fluxes, and as a result the cloud properties (Koren et al., 2004, 2008;
Feingold et al., 2005).
The invigoration mechanism, which refers to larger clouds with larger mass that
develop under polluted conditions was studied mainly in deep convective clouds
(Andreae et al., 2004; Koren et al., 2005; Rosenfeld et al., 2008; Tao et al., 2012; Fan
et al., 2013; Altaratz et al., 2014). Our focus here is on warm cloud fields for which
previous observational studies reported on invigoration effect or a non-monotonic
response of the clouds to an increase in aerosol loading. For example, Kaufman et al.,
(2005) found an increase in cloud fraction (CF) of warm cloud fields with increasing
aerosol loading over the tropical Atlantic Ocean. Yuan et al. (2011) reported that an
increase in volcanic aerosols near Hawaii led to increased trade cumulus CF and
clouds top height. Dey et al. (2011) have shown that an increase in aerosol optical
depth (AOD) from clean to slightly polluted resulted in an increase in CF in warm
clouds over the Indian Ocean. Additional increase in the AOD resulted in a decrease
of CF, explained by the semi direct effect of absorbing aerosols. Costantino and Bréon





(2013) reported higher CF over the south-eastern Atlantic under high aerosol loading
conditions. Koren et al. (2014) have shown that warm convective clouds over the
Southern Oceans can be considered as aerosol limited up to moderate aerosol loading
conditions. As the AOD increases, the clouds were shown to be larger and to produce
stronger rain rates. A reversal in trend of liquid water path (LWP) as a function of
increasing AOD was reported using observations of warm convective clouds under
large range of meteorological conditions (Savane et al., 2015). Li et al. (2011) studied
warm clouds over the southern great plains of the United States and reported no
aerosol effect on clouds' top height.
On the other hand, numerical studies of the aerosol's effect on warm cumulus cloud
fields show either no effect or cloud suppression (meaning shallower and smaller
clouds under higher aerosol loading conditions). Jiang and Feingold (2006) found that
the LWP, CF, and cloud depth of warm shallow convective clouds are insensitive to
an increase in aerosol loading. However, they did demonstrate rain suppression by
aerosols. Xue et al. (2008) showed smaller clouds and suppression of precipitation in
increased aerosol loading environment. Jiang et al. (2010) found a non-monotonic
change in the derivative of the surface rain rate with aerosol loading (susceptibility)
for higher maximal LWP clouds, but a monotonic decrease in the total precipitation
with aerosol loading. Seigel (2014) showed that the clouds' size decreases with
aerosol loading due to enhanced entrainment at clouds' margins.
Some previous studies have demonstrated clouds alteration of their environment
(Zhao and Austin, 2005; Heus and Jonker, 2008; Malkus, 1954; Lee et al., 2014;
Zuidema et al., 2012; Roesner et al., 1990). One example of such effect is the
"preconditioning" or "cloud deepening" effect (Nitta and Esbensen, 1974; Roesner et
al., 1990; Stevens, 2007; Stevens and Seifert, 2008), where clouds cool and moisten
the upper cloudy and inversion layers and by that encourage the development of the
next generation of clouds that encounter improved environmental conditions. This
effect is influenced by the clouds' microphysical properties (Stevens and Feingold,
2009; Saleeby et al., 2015). The role of warm convective clouds in moistening of the
free troposphere was studied intensively using both observations and cloud field
numerical models (Brown and Zhang, 1997; Johnson et al., 1999; Takemi et al., 2004;
Kuang and Bretherton, 2006; Holloway and Neelin, 2009; Waite and Khouider,

125 2010).



Albrecht (1993) used a theoretical single column model to study the effect of
precipitation on the thermodynamic structure of trade wind boundary layer and found
that even low rain rates can dramatically affect the profiles. Under precipitating
conditions, the cloud layer is warmer, drier, and more stable than under non-
precipitation conditions. He also showed that under non-precipitating conditions the
inversion height is greater than under precipitating conditions, due to the larger
amount of liquid water evaporated at those elevations.
Another way clouds effect their environment is by evaporation of rain below the cloud
base which induces cooling of the sub-cloudy layer (Zuidema et al., 2012; Heiblum et
al., 2016a). Lee et al. (2014) demonstrated the aerosol effects on the field's CAPE (as
distributed above cloud base or below it). The organization of the field is influenced
by cloud processes as well. Enhanced evaporative cooling in the sub-cloud layer, for
example, can produce cold pools which enhance the generation of clouds only at their
boundaries, and hence change the organization of the field (Seigel, 2014; Seifert and
Heus, 2013; Heiblum et al., 2016a).
A recent paper (Dagan et al., 2016) showed that polluted clouds act to increase the
thermodynamic instability with time, while clean clouds consume the atmospheric
instability. The trend of the pollution driven increase in the instability is halted once
the clouds are thick enough to develop significant precipitation. Indeed, studies of
long simulation times (>30 hr), showed that the initial differences between clean and
polluted cases are reduced by negative feedbacks of the clouds on the thermodynamic
conditions (Lee et al., 2012; Seifert et al., 2015).
In this work we explore the coupled microphysical-dynamic system of warm marine
cloud fields. We studied how changes in aerosol concentrations affect cloud
properties and the related modifications of the environmental thermodynamic
conditions over time.

### 2. Methodology

The SAM (System for Atmospheric Modeling), non-hydrostatic, anelastic LES model
version 6.10.3 (Khairoutdinov and Randall, 2003) was used to simulate the well-
studied trade cumulus case of BOMEX (Holland and Rasmusson, 1973; Siebesma et
al., 2003). The BOMEX case is an idealized trade-cumulus cloud field that is based on





observations made near Barbados during June 1969. This case was initialized using
the setup specified in (Siebesma et al., 2003). The setup includes surface fluxes and
large scale forcing (see details in Heiblum et al., 2016b). The horizontal resolution
was set to 100 m while the vertical resolution was set to 40 m. The domain size was
12.8 x 12.8 x 4.0 $km^3$ and the time step was 1 sec. Due to computational limitations,
we had to restrict the domain size to a scale that has a limited capacity for capturing
large scale organization (Seifert and Heus, 2013). The model ran for sixteen hours and
the statistical analysis included all but the first two hours (total of 14 hours).
A bin microphysical scheme (Khain and Pokrovsky, 2004) was used. The scheme
solves warm microphysical processes, including droplet nucleation, diffusional
growth, collision coalescence, sedimentation and breakup.
The aerosol distribution was based on measurements of marine aerosol size
distribution (see details in Jaenicke 1988 and Altaratz et al., 2008). Eight different
simulations were conducted with a changing aerosol concentration (5, 25, 50, 100,
250, 500, 2000 and 5000 $cm^{-3}$ (Dagan et al., 2015a).


## 3. Results and discussion

### 3.1 Mean cloud field properties under different aerosol loading conditions

The aerosol effects on the mean properties of the eight simulated cloud fields are
examined first. Figure 1 presents mean values of key properties of cloud fields as a
function of the aerosol loading for the entire (14 h) simulation time.
The total water mass (calculated as mean over time in each domain) as a function of
aerosol concentration shows a clear reversal in the trend (Fig. 1A). It increases when
increasing aerosol loading from 5 to 50 $cm^{-3}$. Additional increase in the aerosol
loading results in a decrease in the total water mass in the domain.
The LWP (Liquid Water Path - Fig. 1B) calculated as a mean over time over all
cloudy columns in each domain, (which is strongly correlated with the total water
mass), also shows the same non-monotonic general trend. The maximum in the curve
of cloudy LWP is at slightly higher aerosol concentration compared to the total mass
(100 $cm^{-3}$). This difference can be explained by the link to the cloud fraction (CF –
calculated as the area covered by clouds with optical path $\tau > 0.3$ Fig. 1C) that
decreases above aerosol loading of 25 $cm^{-3}$. And so, for the more polluted simulations





the mass is distributed on smaller horizontal cloud areas as shown in previous studies
(Seigel, 2014).

There is also a significant difference in the way the water mass is distributed along the
atmospheric column in the different simulations. The maximum cloud top height (Fig.
1D), calculated as a mean over time of the altitude of the highest grid box in the
domain that contains liquid water content (LWC >0.01g/kg) increases significantly
when increasing aerosol loading up to 500 cm$^{-3}$ (increase from 1692 m to 2120 m
when increasing aerosol loading from 5 to 500 cm$^{-3}$). Additional increase in the
aerosol loading results in a minor decrease in the maximum cloud top height (down to
2030 m for aerosol loading of 5000 cm$^{-3}$). The minor decrease seen for this range of
aerosol concentration (compared with the larger decrease in the mean LWP for
example) can be explained by the location of the maximal cloud top height above the
cloud core, which is affected mainly by the invigoration processes (enhanced
condensation and latent heat release) and less by margin oriented processes (enhanced
entrainment and evaporation) that significantly impact the total cloud mass (Dagan et
al., 2015a). Another reason is the cloud deepening effect under polluted conditions
(Stevens, 2007; Seifert et al., 2015) that will be described later. As for the mean cloud
top height calculated as a mean of all cloudy columns along the whole run (Fig. 1E),
the trend shows a monotonic increase with aerosol loading. The trend is approaching a
saturation level for high aerosol concentration values. The mean cloud top value over
the simulation is 810 and 1010 m for the simulations with aerosol loading of 5 to 5000
cm$^{-3}$, respectively.

The trend in the domain's average rain rate, as a function of the aerosol loading (Fig.
1F) shows a peak at relatively low aerosol loading (similar to optimal value of the CF)
of 25 cm$^{-3}$.

Fig. 2 presents the mean vertical profiles of the condensation-evaporation tendencies,
for four different simulations. We note that as the aerosol loading increases, both the
mean condensation and evaporation rates increase (Dagan et al., 2015a; Koren et al.,
2014; Pinsky et al., 2013; Seiki and Nakajima, 2014). Below cloud base (located



around 550 m) the clean simulations have small rain evaporation values which is
absent in the polluted simulations.

Effective terminal velocity ($\eta$) is defined as the mass weighted average terminal
velocity of all the hydrometeors within a given volume of air (Koren et al., 2015). By
definition, $\eta$ measures the terminal velocity of the water mass's center of gravity
(COG), i.e. the COG's movement with respect to the surrounding air's vertical
velocity (W). Small absolute values $|\eta|$ imply that the droplets COG will move better
with the surrounding air, i.e. the droplets will have better mobility (Koren et al.,
2015). The sum $V_{COG} = W + \eta$ ($\eta$ always negative) reflects the water mass COG
vertical velocity relative to the surface. Positive $V_{COG}$ implies a rise of the COG, and
negative value means falling.
The mean updraft (in both space and time, weighted by the liquid water mass in each
grid box to be consistent with the COG point of view - Fig. 3A) increases with the
increase in aerosol loading, in agreement with previous studies (Saleeby et al., 2015;
Seigel, 2014). This indicates an increase in the latent heat contribution to the cloud
buoyancy, driven by increase in the condensation efficiency (Dagan et al., 2015a,b;
Koren et al., 2014; Pinsky et al., 2013; Seiki and Nakajima, 2014) (Fig. 2). At the
same time, $|\eta|$ decreases as the aerosol concentration increases (Fig. 3B) indicating
better mobility of the smaller droplets, allowing them to move more easily with the
air's updrafts. The outcome of these two effects is an increased $V_{COG}$ for higher
aerosol concentration (Fig. 3C) indicating that the polluted clouds' liquid water is
pushed higher in the atmosphere (Koren et al., 2015) as shown by higher COG (Fig.
3D).

The mean COG height of the water mass (Grabowski et al., 2006; Koren et al., 2009)
(Fig. 3D), increases with the aerosol loading up to a relatively high concentration (500
$cm^{-3}$). Note that while the trend in the system's characteristic velocities ($\eta$ and W) is
monotonic increase, the COG has an optimal aerosol concentration for which it
reaches its maximum height (500 $cm^{-3}$). For aerosol concentrations above 500 $cm^{-3}$ a
minor decrease is shown. As described above, the COG height increase with aerosol





loading, between extremely clean and polluted conditions, can be explained by
increased $V_{COG}$, which is a product of both lower $|\eta|$ and increased updraft in the
cloud scale, and larger thermodynamic instability induced by the polluted clouds in
the field scale as will be shown in the next section (Dagan et al., 2016; Heiblum et al.,
2016a). The reduction of the mean COG height in the most polluted simulations is
caused by cloud suppressing processes including an enhanced entrainment (see the
enhanced evaporation efficiency with aerosol loading – Fig. 2) and larger water
loading (Dagan et al., 2015a - shown also in Fig. 4 below).
The trend in COG height can be also viewed (in more detail) in Fig. 4 that presents
profiles of mean LWC.
We show that both the height and the magnitude of the maximum LWC increase with
the aerosol loading. Below the clouds' base (H<~550m) the LWC trend is reversed
due to the enhancement of rain in the clean runs (Fig. 1F). The increase in LWC with
aerosol loading implies a larger water loading negative component in the clouds'
buoyancy.

All the evidence presented in Figs. 2-4 explains the non-monotonic trends of the
clouds properties response to changes in aerosol loading (Fig. 1). For clean conditions
(below the optimal aerosol concentration value), an increase in aerosol loading would
enhance the cloud development (larger mass, LWP, cloud top, CF, rain rate) because
of two main factors: 1) an increase in the condensation efficiency (due to the larger
total droplet surface area for condensation and longer time- Fig. 2), and 2) smaller
effective terminal velocity ($\eta$) values, that per given updraft allow the cloud's
hydrometeors to be pushed higher in the atmosphere (Koren et al., 2015) (Fig. 3B).
The higher condensation efficiency in polluted clouds (Fig. 2) results in a larger latent
heat release that enhances the updraft (Fig. 3A) and cloud development. The increased
$V_{COG}$ reflects the two cloud enhancing processes (decrease in $|\eta|$ and larger mean
updraft). We note that the increase in the mean updraft values with aerosol loading is
seen despite the negative effect of water loading (see Fig. 4). For aerosol
concentrations above the optimum, cloud development is suppressed by the increase
in evaporation efficiency (Fig. 2) and hence stronger mixing of the cloud with its



environment (i.e. Small et al., 2009), and larger water loading due to rain suppression
(Dagan et al., 2015a, Fig. 4).


### 289    3.2 The time evolution of the mean cloud field properties under different aerosol
### 290    loading conditions

All the aerosol effects that were discussed up to this point (condensation-evaporation
efficiencies, $\eta$ and water loading) are applicable both on the single cloud scale as well
as on the cloud field scale. However, on the cloud field scale, another aspect needs to
be considered, namely the time evolution of the effect of clouds on the field's
thermodynamic conditions.
Figure 5 presents the changes (final value less initial one) in the temperature (T) and
water vapor content ($q_v$) vertical profiles as a function of aerosol concentration used in
the simulation. The initial profiles were identical in all simulations. In low aerosol
concentration runs (100 cm$^{-3}$ and below) the sub-cloud layer becomes cooler and
wetter with time and the cloudy layer warmer and drier. Meanwhile, under higher
aerosol concentrations conditions (250 cm$^{-3}$ and above) the sub-cloud layer becomes
warmer and drier while the cloudy and inversion layers become colder and wetter.
This trend is driven by the condensation-evaporation tendencies along the vertical
profile (see Fig. 2, Dagan et al., 2016). Under low aerosol concentration conditions,
water condenses at the cloudy layer and is advected downward to the sub-cloud layer
where it partially evaporates. Under polluted conditions, on the other hand, the
condensed water from the lower part of the cloudy layer is advected up to the upper
cloudy and inversion layers (driven by larger $V_{COG}$ - Fig. 3) and evaporates there
(Dagan et al., 2016).

Such trends in the environmental thermodynamic conditions are likely to affect the
forming clouds. In Fig. 6 the time evolution of some of the key cloud field properties
are considered (the same properties that were shown in Fig. 1). The blue, green and
red curves represent the mean values over the first, second and third periods of the
simulations, respectively (each one covers 4 hours and 40 min). Table 1 presents





change (in percentage) in the mean values of key variables between the third period of
the 8 simulations (during the 11:20-16:00 hours of simulation, red curves in Fig. 6)
and the first period (02:00-06:40 hours of simulation, blue curves in Fig. 6).
Examination of the evolution in the mean total water mass along the simulations (Fig.
6A blue, green and red curves) presents a different trend between the clean and the
polluted simulations. In the clean simulations (5-100 cm$^{-3}$) the total water mass
decreases significantly with time (a decrease of 57, 45, 44, 20% in the total mass for
the cases of 5, 25, 50 and 100 cm$^{-3}$ respectively – see table 1). On the other hand, in
the more polluted simulations, (with aerosol loading of 250 and 500 cm$^{-3}$) there is an
increase in the total water mass with time (of 17 and 37% between the first and the
last third periods of the simulations, respectively). Under extreme polluted conditions
of 2000 and 5000 cm$^{-3}$, the total water mass in the domain is small and there is little
change with time. These changes in time push the optimum aerosol concentration to
higher values along the simulation time. This trend is also shown for the optimum
aerosol concentration with regard to the mean cloudy LWP (Fig. 6B), max top (Fig.
6D) and mean top (Fig. 6E).
Trends in the mean rain rate show that in the cleanest simulations (5, 25 and 50 cm$^{-3}$)
it decreases with time (Fig. 1H, 53.3, 32.9 and 40.1%, respectively). In the regime of
medium to fairly high aerosol loading (100, 250 and 500 cm$^{-3}$) the rain rate increases
(19.6, 598.1 and 841.5%, respectively). And in the most polluted simulations (2000
and 5000 cm$^{-3}$) the surface rain is negligible throughout the simulation time.

The time evolution of the thermodynamic conditions (Fig. 5) shows a reduction
(enhancement) in the thermodynamic instability with time in the clean (polluted)
simulations. Figure 6 and table 1 indicate that under clean conditions the decrease in
the thermodynamic instability with time leads to a decrease in the mean cloud field
properties such as total mass and cloud top height. Under polluted conditions the
trends are opposite and the mean cloud field properties increase with time due to the
increase in thermodynamic instability (Dagan et al., 2016) and due to the cloud
deepening (Stevens and Seifert, 2008; Stevens, 2007; Seifert et al., 2015). These
differences between the clean and polluted simulations drive changes in the optimum
aerosol concentration with time. For example, for the LWP (Fig. 1B) the optimum



aerosol concentration is 50, 100 and 250 $cm^{-3}$ for the first, second and third parts of
the simulation, respectively.


## **Summary**

Cloud processes can be divided in a simplistic manner into two characteristic scales –
the cloud scale and the field scale. Here using LES model with bin microphysical
scheme we studied the outcome of the two scales' processes acting together. We first
presented domain averaged properties over the whole simulation time (section 3.1) to
indicate the general aerosol effects in a first order manner and then we followed the
time evolution of the effects (section 3.2).
A non-monotonic aerosol effect was reported recently for a single cloud scale (Dagan
et al., 2015a,b). Here we show that these trends "survived" the domain and time
averaging. We argue that the enhanced development branch trend is driven by two
main processes of enhanced condensation and reduced effective terminal velocity
(which improves the droplets mobility). These processes are mainly related to the core
of the clouds and to the early stages of clouds development. We show that the cloud's
systems characteristic velocities can capture these effects. The effective terminal
velocity ($\eta$) inversely measures the mobility. Smaller droplets with smaller variance
will have smaller $\eta$ and therefore will be pushed higher in a given updraft, whereas
larger droplets with larger $\eta$ will deviate downward faster from the surrounding air.
Increase in condensation efficiency drives more latent heat release that enhances the
cloud updraft. We showed that $V_{COG}$ is a product of the two velocities.
The descending branch in which increase of aerosol loading suppresses cloud
development is governed by increase in the evaporation efficiency on the subsaturated
parts of the clouds and by increase in water loading.
Since clouds change the atmospheric thermodynamic conditions in which they form,
different initial clouds would cause different impact on the environment. Therefore,
cloud field is a continuously evolving system for which aerosol properties determine
an important part of the temporal trends. Figure 5 shows striking differences between
the evolution of the thermodynamic profiles in clean and polluted cases. For the
polluted clouds (mostly non-precipitating), the upper cloudy layer turns wetter and



cooler due to enhanced evaporation and the sub-cloudy layer becomes warmer and
drier, which altogether act to increase the instability. On the other hand, clean
precipitating clouds consume the initial instability with time by warming the cloudy
layer (due to latent heat release) and cooling the sub-cloud layer by evaporation of
rain.
The polluted cloud feedbacks on the thermodynamic conditions act to deepen the
clouds. Since clouds that form in a more unstable environment are expected to be
aerosol limited up to higher aerosol concentrations (Koren et al., 2014; Dagan et al.,
2015a), an increase in the domains instability for the polluted cases drives an increase
in the optimal aerosol concentration with time.
We note that such an increase in the instability cannot last forever. A deepened cloud
will eventually produce larger precipitation rates that may weaken the overall effect
on the field (Stevens and Feingold, 2009; Seifert et al., 2015). These results pose an
interesting question on the dynamical state of cloud fields in nature. Do the cloud
fields 'manage' to reach a "near-equilibrium" state (Seifert et al., 2015), for which the
deepening effect balances the aerosol effect fast enough that the effects are buffered
most of the time (Stevens and Feingold, 2009). Or maybe, the characteristic lifetime
of a trade cumulus cloud field is shorter than the time it takes to significantly balance
the aerosol effects. In this case the cloud fields could be regarded as 'transient' and
therefore, as shown here, aerosol might have a strong effect on the clouds, both
through affecting the microphysics, initiating many feedbacks in the cloud scale, and
by affecting the field thermodynamic evolution over time.

## Acknowledgements

This research has been supported by the Minerva foundation with funding from the
Federal German Ministry of Education and Research.

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



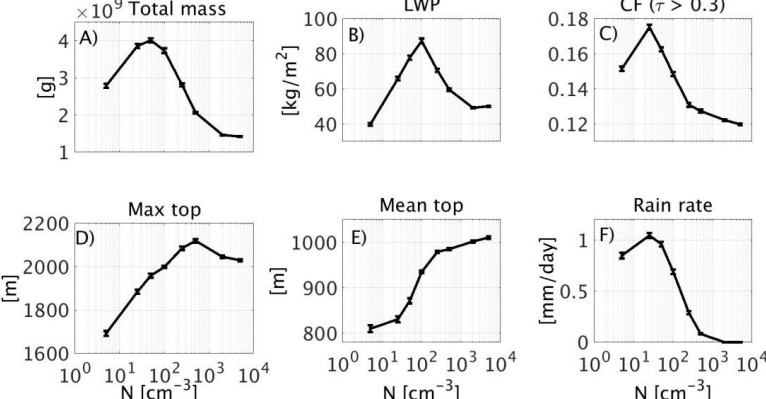

**Figure 1. mean properties (over domain and time) of the simulated cloud fields as a function of the aerosol concentration used in the simulation: A) total liquid water mass in the domain, B) cloudy LWP, C) cloud fraction (CF) for columns with τ>0.3, D) maximum cloud top, E) mean cloud top, and, F) surface rain rate. Each of these mean properties are calculated for the last 14 hours out of the 16 hours of simulation. The error bars present the standard errors. For details about the different properties see the text.**

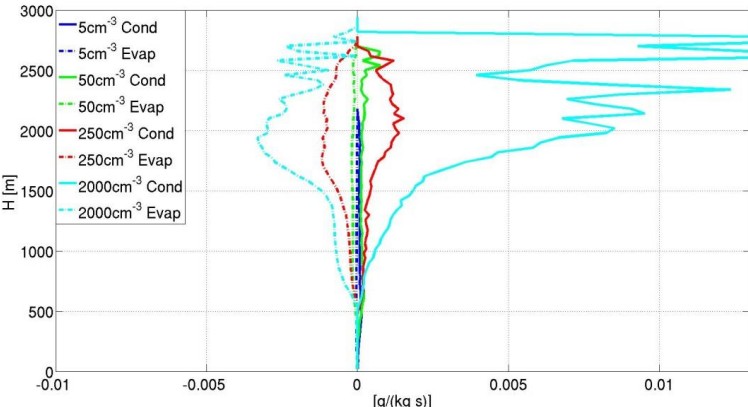

**Figure 2. Domain mean condensation (solid lines) and evaporation (dashed lines) tendencies for four different simulations conducted with different aerosol concentration levels (5 cm⁻³ blue, 50 cm⁻³ green, 250 cm⁻³ red and 2000 cm⁻³ cyan).**



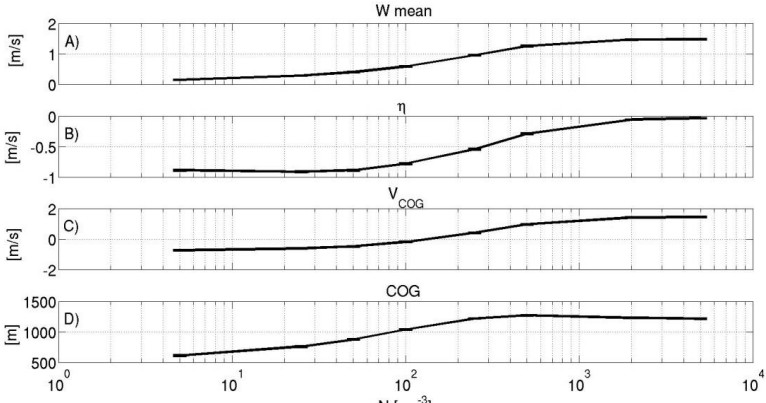


**Figure 3. Mean (over time and space) of A) updraft (W), B) effective terminal velocity ($\eta$), C) the center of gravity velocity $V_{COG}$ and D) COG (center of gravity) height as a function of the aerosol concentration. All calculated for the last 14 hours out of the 16 hours of simulation.**



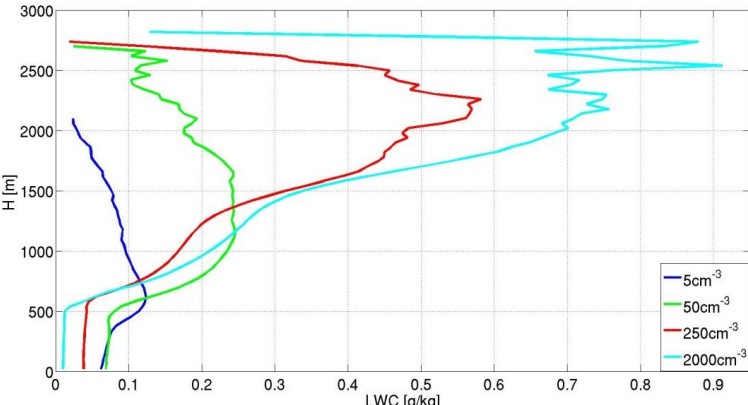


**Figure 4. Mean liquid water content (LWC) vertical profiles for four different simulations (5 cm$^{-3}$ blue, 50 cm$^{-3}$ green, 250 cm$^{-3}$ red and 2000 cm$^{-3}$ cyan). The mean profiles are calculated for the last 14 hours out of the 16 hours of simulation.**







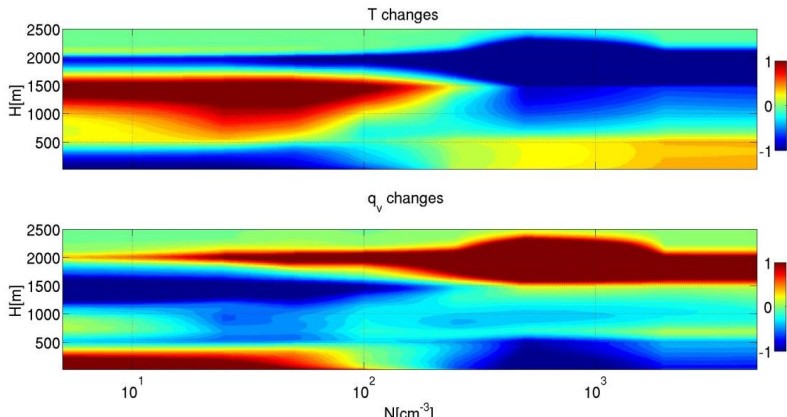

**Figure 5. Total change, during 16 h of simulation in the temperature ([k] upper panel) and water vapor content ([g/kg] – lower panel) domain mean vertical profiles as a function of the aerosol concentration used in the simulation.**

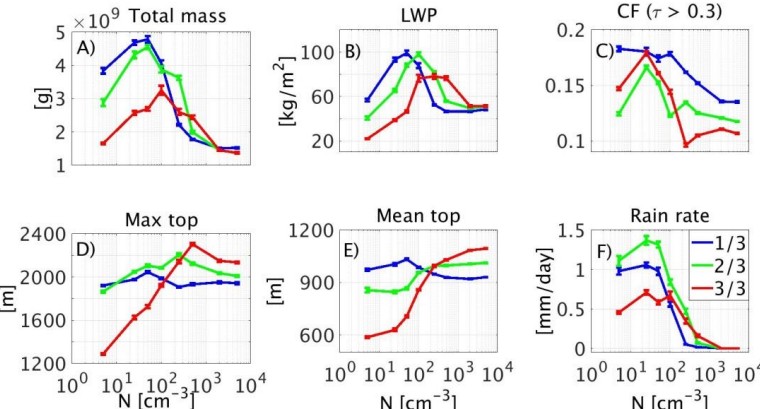

**Figure 6. Mean properties (over time and domain) of the simulated cloud fields as a function of the aerosol concentration used in the simulation: A) total liquid water mass in the domain, B) cloudy LWP, C) cloud fraction (CF) for columns with $\tau > 0.3$, D) maximum cloud top, E) mean cloud top, and, F) surface rain rate. Each property is calculated separately for each period of one third of the simulations (blue, green and red for the first, second and third periods, respectivly). The error bars present the standard error. For details about the different properties, see the text.**





**Table 1. change (in %) in key variables between the mean values in the last third period of the**
**simulations and the first period. Negative values are presented in red.**

|  | Total mass [%] | LWP [%] | COG [%] | Max top [%] | Mean top [%] | W max [%] | CF [%] | Rain rate [%] |
|---|---|---|---|---|---|---|---|---|
| 5 cm$^{-3}$ | -57.0 | -61.4 | -43.1 | -32.9 | -39.7 | -28.2 | -19.7 | -53.5 |
| 25 cm$^{-3}$ | -45.2 | -58.3 | -39.6 | -17.8 | -37.4 | -38.8 | -0.6 | -32.9 |
| 50 cm$^{-3}$ | -43.8 | -53.1 | -33.7 | -15.6 | -31.6 | -47.9 | -7.5 | -40.1 |
| 100 cm$^{-3}$ | -20.1 | -13.0 | -16.1 | -3.2 | -13.0 | -32.8 | -19.0 | 19.6 |
| 250 cm$^{-3}$ | 17.5 | 48.6 | 5.0 | 12.4 | 5.0 | -4.3 | -40.7 | 598.1 |
| 500 cm$^{-3}$ | 37.4 | 64.2 | 19.9 | 19.2 | 10.7 | 9.4 | -30.9 | 841.5 |
| 2000 cm$^{-3}$ | -3.7 | 10.6 | 14.8 | 10.1 | 17.9 | 6.0 | -17.8 | - |
| 5000 cm$^{-3}$ | -10.1 | 5.7 | 13.7 | 9.9 | 17.5 | 2.9 | -20.7 | - |

