# Peer review of "changes in aerosol loading"

_Atmospheric Chemistry and Physics, 2016_

## Referee Comment (RC1) · Anonymous Referee #1 · 22 Sep 2016

General Comments:

This article investigated the response of thermodynamic properties of cloud fields to changes in aerosol loading, using aLarge Eddy Simulations (LES) with bin microphysics. The results that pollution acts to suppress rain andincrease atmospheric instability, that is, warming of the lower part of the cloudy layer and cooling of the upper part, are very important and add some new insights into the understanding of aerosol-cloud-radiation interactions. The article is generally well written, concise and should be publishable if the following specific comments and suggestions can be considered in revision.

Specific Comments:

[Figure]

1. Since I did not see the article by Dagan et al. (2016), but from the title and introduction in this manuscript, it seems to me that the results and conclusions of these two paper are similar. What are the main differences between them?

2. Different initial concentrations of aerosol particles are used in the simulation. How the initial aerosols are distributed vertically, uniform or decrease according to a certain function? Whether they change with time? What are the altitudes of these aerosol concentrations referred to? Did you consider aerosol regeneration after evaporation of cloud particles? This could be avery important source of aerosols, especially in polluted conditions, and could be of important effects to the subsequently developed clouds and precipitation (e.g., Yin et al. 2005).

3. Whether the reversing point (line 182) change with thermodynamic and dynamic conditions?

4. Some of the results (Line 189-190, 201)for more polluted simulations contradict with the Twomey effects. Is there any observational evidence to support these results? 5. Line 198-200: Is the invigoration effect limited to aerosol concentration lower than 500 cm-3?

6. Line 251-252: Similar trend is also seen for maximum cloud top height. Is the decrease in COG height for larger aerosol concentration related to the inversion layer above cloud which prohibited the further growth of clouds?

7. Line 259-260: The LWP is decreasing with larger aerosol concentration. Is the water loading larger?

8. Line 297-299: Suggest to show 1-2 figures related the time variations of cloud fields to support the statements.

9. Line 331-335: Suggest to add more explanations to the results.

Technical corrections:

1. Line 35: add "water vapor and" at the end of this line;

2. Line 63: remove ","between the parentheses;

3. Line 158: "in (Siebesma et al., 2003)"should be replaced by "by Siebesma et al. (2003)";

4. Line 184-185: remove ();

5. Line 295: change "less" to "minus";

6. Line 625: Add the variable for the abscissa.

References:

Yin, Y., K. S. Carslaw, and G. Feingold, 2005: Vertical transport and processing of aerosols in mixed-phase convective cloud and the feedback on cloud development. Q. J. R. Meteorol. Soc., 131, 221-246.

---

## Referee Comment (RC2) · Anonymous Referee #2 · 22 Sep 2016

(MS No.: acp-2016-736)

Comments to the Author:

The manuscript investigated cloud-aerosol interaction in warm cloud environment using a Large Eddy Simulations (LES) with bin microphysics. The paper is generally well written, contains some potentially interesting model results, but does not go into sufficient detail in any one of the cases and there is no comparison with observations which is necessary for a reality check. The article can be considered for publication after major revision.

[Figure]

Some specific comments and suggestions follow:

1. The algorithm of cloud-aerosol interaction in the model is not clear. Suggest to elaborate the section 2 (methodology) in details. How the aerosol initiations are taking place? What is the composition of aerosols they have considered in their model? 2. Does the model take nucleation scavenging? Need to be cleared. 3. How does different size distribution of aerosols (CCN) affect the DSD? What is the effect of large CCN on the processes of droplet nucleation? 4. Authors have discussed on the role of aerosol on collision-coalescence, delayed surface rain in their introduction (line 53-56) and also about cloud invigoration mechanism in line 80-83. In this context authors should put the references (Hazra et al. 2013a, Journal of Atmospheric Science; Hazra et al. 2013b, Journal of Geophysical Research-Atmosphere) studied cloud-aerosol interactions using 2 moment bulk microphysical scheme (Cheng et al. 2007, QJRMS) in meso-scale model. In this regard, I suggest refer/include all those papers in the manuscript. 5. Line 87-91: How aerosol impact on cloud fraction (CF), discussion not clear. Line: 169-172: Authors have mentioned the aerosols number concentration. Is it total concentration of all bins? Or which bin size of aerosols number concentrations they are increasing. This should be cleared to the readers. 6. Line 177-179: Authors mentioned "the eight simulated cloud fields are examined first". Are the eight simulated cloud fields are in similar or different. Need to be explained. 7. In Figure 1 and Figure 6: Total liquid water mass, LWP, CF and rain rate show a "tipping" point. It will be worthy if authors discuss on the "tipping point" as revealed in Figure 1 and 6. Another important point authors should explain why the "tipping point" for total water mass, LWP are different from the CF and rain rate. 8. Can authors consider another aerosol concentration at lower rage? Authors have showed the response of vertical wind (updraft) to varying CCN concentrations (Figure 3A). 9. It shows updraft is monotonically increasing with aerosol concentrations. Is it correct? Need to be explained. 10. Another doubt: Figure 1B and Figure 4: Figure 1B shows clear tipping point of LWP, first it increases then it decreases, whereas when I am looking into the Figure 4, LWC increasing with aerosol number concentrations. Authors should make it unambiguous. 11. Time dependent

part (Figure 6, line 338-349) is not much convincing. Need to be explained in details. 12. It will be worthy if authors provide cloud drop number concentration and cloud drop size with increasing aerosol concentration. 13. Can authors put some hints of cloud-aerosol interactions in the mixed-phase clouds in their "Conclusion" section? 14. Comments Line 35-36: It is not only depends on CCN concentrations, also depends on availability of water vapor. 15. The authors probably should proof read the manuscript carefully to correct some typos.

——————————————————— References: Cheng CT, Wang WC, Chen JP. 2007. A modeling study of aerosol impacts on cloud microphysics and radiative properties. Quart. J. Roy. Meteor. Soc., 133: 283-297. Hazra A, Goswami BN, Chen JP. 2013a. Role of Interactions between Aerosol Radiative Effect, Dynamics and Cloud Microphysics on Transitions of Monsoon Intraseasonal Oscillations. Journal of Atmospheric Sciences, 70: 2073-2087, DOI: 10.1175/JAS-D-12-0179.1, 2073-2087. Hazra A., Mukhopadhyay P., Taraphdar S., Chen J-P., Cotton W.R. 2013b. Impact of aerosols on tropical cyclones: An investigation using convection-permitting model simulation. Journal of Geophysical Research, 118, 7157–7168, DOI:10.1002/jgrd.50546.

---

## Author Comment (AC1) · 15 Nov 2016

**Replay to the review of: "Time dependent, non-monotonic response of warm convective cloud fields to changes in aerosol loading"**

We would like to thank the reviewers for their efforts and beneficial comments that helped us improve our paper and present a clearer and more complete study. Before addressing all of the reviewers' comments in a point by point manner, we would like to list a few general revisions that are relevant to both reviewers:

1. We added information in the methodology section for describing better the treatment of aerosol in the model including the aerosol size distribution, chemistry, activation etc.

2. Following both of the reviewers' comments we added a supporting information (SI) file with more technical information. The SI includes information about the mean cloud and rain drop concentration and size for the different simulations, vertical profiles of mean cloud fraction and additional information about the time evolution of the thermodynamic conditions.

A point by point reply to all of the reviewers' comments (answers in blue).

**Referee #1**
**General Comments:**

This article investigated the response of thermodynamic properties of cloud fields to changes in aerosol loading, using a Large Eddy Simulations (LES) with bin microphysics. The results that pollution acts to suppress rain and increase atmospheric instability, that is, warming of the lower part of the cloudy layer and cooling of the upper part, are very important and add some new insights into the understanding of aerosol-cloud-radiation interactions. The article is generally well written, concise and should be publishable if the following specific comments and suggestions can be considered in revision.

Answer: we thank the reviewer for finding this work important and for the encouragement.

**Specific Comments:**

1. Since I did not see the article by Dagan et al. (2016), but from the title and introduction in this manuscript, it seems to me that the results and conclusions of these two paper are similar. What are the main differences between them?

   Answer: In this study we examine the response of cloud fields' mean properties to changes in aerosol loading. This is done both globally (during the entire simulation period, Sec. 3.1) and for different parts of the simulation period (Sec. 3.2). We show that the cloud fields' mean properties response in a non-monotonic way to an increase in aerosol loading. The non-monotonic trends with an optimal aerosol concentration of the mean cloud field properties are explained by contradicting aerosol effects. The time evolution of this response and the increase along time of the optimal aerosol concentration are driven by differences in the evolution of the thermodynamic conditions in the cloud field under different aerosol concentrations. This thermodynamic evolution under different aerosol concentrations is the focus of Dagan et al. (2016) paper. In Dagan et al. (2016) we don't discuss the aerosol effect on the mean cloud field properties and their non-monotonic trend but only the thermodynamic evolution. As the changes in the thermodynamic conditions is the driving force of the evolution of the non-monotonic trend, and for the purpose of writing a "stand-alone" paper this issue is explained here again.

   We added clarification to the revised manuscript: "*All the aerosol effects that were discussed up to this point (condensation-evaporation efficiencies, $\eta$ and water loading) are applicable both on the single cloud scale as well as on the cloud field scale. However, on the cloud field scale, another aspect needs to be considered, namely the time evolution of the effect of clouds on the field's thermodynamic conditions (which was the focus of a recent study by Dagan et al., 2016).*"

2. Different initial concentrations of aerosol particles are used in the simulation. How the initial aerosols are distributed vertically, uniform or decrease according to a certain function? Whether they change with time? What are the altitudes of these aerosol concentrations referred to? Did you consider aerosol regeneration after evaporation of cloud particles? This could be a very

important source of aerosols, especially in polluted conditions, and could be of important effects to the subsequently developed clouds and precipitation (e.g., Yin et al. 2005).

Answer: We thank the reviewer for rising this important point. The model is initialized using an oceanic aerosol size distribution (Jaenicke, 1988; Altaratz et al., 2008). The concentrations we report are near the surface level and the aerosols are assumed to maintain constant mixing ratio with height. A constant aerosol mixing ratio is expected in a well mixed environment i.e. in the boundary layer. We note that the focus here is on shallow convective clouds and hence this is a reasonable assumption. Under the assumption of constant aerosol mixing ratio, its concentration decreases with height according to the decrease in the air density. Initially at the inversion base height (~1500m) the aerosol concentration is ~12.3% less than at the surface.

The aerosols serve as potential cloud condensation nuclei (CCN) and undergo activation based on Kohler theory (Khain et al., 2000). A prognostic equation for the aerosol is solved, which includes regeneration upon evaporation and removal by surface rain. The aerosols are assumed to be composed of ammonium-sulfate.

We added clarification to the revised manuscript: "*The shape of the aerosol size distribution was based on measurements of marine aerosol distribution (see details in Jaenicke 1988 and Altaratz et al., 2008). Eight different simulations were conducted with a changing aerosol concentration (total concentration of: 5, 25, 50, 100, 250, 500, 2000 and 5000 $cm^{-3}$ near ground level, Dagan et al., 2015a). For reducing the sensitivity of our results to the aerosol size distribution shape and focusing on the aerosol number concentration effect, the different aerosol concentrations are calculated by multiplication of all bins in the smallest concentration size distribution by a constant factor and maintaining a similar shape of the size distribution. The aerosol is assumed to be composed of ammonium-sulfate and maintains constant mixing ratio with height. A prognostic equation is solved for the aerosol mass, including regeneration upon evaporation and removal by surface rain. Regeneration upon evaporation of cloud drops was shown to be a very important source of aerosols, especially in polluted conditions (Yin et al., 2005). The aerosol serves as potential cloud condensation nuclei (CCN)*

*and it is activated based on the Kohler theory (Khain et al., 2000). The aerosol (water drop) size distribution is calculated between 5 nm to 2 μm (2 μm-3.2 mm). For both aerosol and drops, successive bins represent doubling of the mass."*

3. Whether the reversing point (line 182) change with thermodynamic and dynamic conditions?

   Answer: we thank the reviewer for this comment. The reversing point (in this case for the trend of the maximum water mass as a function of aerosol loading) does change with the thermodynamic conditions (Dagan et al., 2015a). The interplay between the different aerosol effects depends on the environmental conditions. For example, under dryer conditions or smaller clouds the enhanced evaporation at the cloud margin and the consequence mixing with the outside dryer air under polluted conditions will be more significant than under humid conditions and for larger clouds (Dagan et al., 2015a).

   We added clarification to the revised manuscript: *"The total water mass (calculated as mean over time in each domain) as a function of aerosol concentration shows a clear reversal in the trend (Fig. 1A). For the given environmental conditions simulated here, it increases when increasing aerosol loading from 5 to 50 cm$^{-3}$. Additional increase in the aerosol loading results in a decrease in the total water mass in the domain."*

   In addition, we added clarification about the dependency of the optimal aerosol concentration on the environmental conditions to the introduction of the revised manuscript: *"Aerosol effects on single warm convective clouds were shown to have an optimal value with respect to maximal water mass, cloud depth and rain yield (Dagan et al., 2015a,b), which depends on the environmental conditions."*

   In Sec. 3.2 of the manuscript, and especially in Fig. 6, we show how, in our case, the optimal aerosol concentration changes with the thermodynamic conditions along the simulations: *"These changes in time push the optimum aerosol concentration to higher values along the simulation time."*

4. Some of the results (Line 189-190, 201) for more polluted simulations contradict with the Twomey effects. Is there any observational evidence to support these results?

Answer: according to Twomey effect (Twomey, 1974; Twomey, 1977), for a given liquid water path (LWP) aerosol act to reduce the droplets size and hence increase the cloud albedo. Figure S1 below presents the mean vertical profiles of the droplet radius and concentration for four different simulations differ by the aerosol concentration. It demonstrates, in agreement with Twomey effect, that the mean cloud droplet radius decreases with aerosol loading (in the cloudy layer H>500m). Below cloud base the trend is reversed and the rain drops size increases with the aerosol loading (Altaratz et al., 2008; Berg et al., 2008). Based on this figure we believe that our results do not contradict the Twomey effect.

In lines 189-190 of the manuscript we describe the results presented in Fig. 1C of the manuscript of a decrease in cloud fraction (calculated as the area covered by clouds with optical path $\tau$>0.3) with aerosol loading above 25 cm$^{-3}$. This was shown before based both on observational (Small et al., 2009) and numerical studies (Seigel, 2014; Xue and Feingold, 2006; Xue et al., 2008). In line 201 we mention that additional increase in the aerosol loading above 500 cm$^{-3}$ results in a minor decrease in the cloud top height which is also related to the enhanced evaporation at the cloud margin with aerosol loading (Small et al., 2009) and does not contradict the Twomey effect.

For clarifying this issue and providing more information about the aerosol effect on the clouds mean properties in our simulations we have added the figure below to the SI (Fig. S1).

The additions to the revised manuscript: *"The effects of changes in aerosol concentration on the drop concentration and its mean size, for the different simulations can be found in Fig. S1 in the supporting information (SI)."*

The additions to the SI: "*Figure S1 presents vertical profiles of the mean concentration and mean cloud drop size per height. It demonstrates that at the cloudy layer (H>500m) the mean drop size decreases with aerosol loading, while its concentration increases (Twomey, 1977). Below cloud base the trend*

*is reversed – larger rain drops and lower concentration under more polluted conditions (similar to what was shown in Altaratz et al., 2008)."*

[Figure]

*Figure S1. Vertical profiles of a) the mean (over time and domain) drop radius, and b) the mean (over time) of drop maximum concentration. These results include both cloud and raindrops for four simulations (with aerosol loading levels of 5 cm$^{-3}$ – blue, 50 cm$^{-3}$ – green, 250 cm$^{-3}$ – red, and 2000 cm$^{-3}$ – cyan).*

5. Line 198-200: Is the invigoration effect limited to aerosol concentration lower than 500 cm$^{-3}$?

   Answer: For the maximum cloud top height and the case simulated here (BOMEX case study) the reversal in the trend occurs at aerosol concentration of 500 cm$^{-3}$ (we expect this value to be higher for deeper clouds). Above this concertation the negative effects of aerosols, namely the increase in evaporation efficiency at the cloud margin and the water loading take over and dominate over the positive effects of increased condensation efficiency at the cloud core and the decrease in effective terminal velocity. The optimal aerosol concentration for the maximum cloud top height is higher than that of the liquid water mass (Dagan et al., 2015a) because it is affected more by the positive aerosol effects in the cloud core. As we explain it in the text: *"The maximum cloud top height (Fig. 1D), calculated as a mean over time of the*

*altitude of the highest grid box in the domain that contains liquid water content (LWC >0.01g/kg) increases significantly when increasing aerosol loading up to 500 cm⁻³ (increase from 1692 m to 2120 m when increasing aerosol loading from 5 to 500 cm⁻³). Additional increase in the aerosol loading results in a minor decrease in the maximum cloud top height (down to 2030 m for aerosol loading of 5000 cm⁻³). The minor decrease seen for this range of aerosol concentration (compared with the larger decrease in the mean LWP for example) can be explained by the location of the maximal cloud top height above the cloud core, which is affected mainly by the invigoration processes (enhanced condensation and latent heat release) and less by margin oriented processes (enhanced entrainment and evaporation) that significantly impact the total cloud mass (Dagan et al., 2015a). Another reason is the cloud deepening effect under polluted conditions (Stevens, 2007; Seifert et al., 2015) that will be described later."*

6. Line 251-252: Similar trend is also seen for maximum cloud top height. Is the decrease in COG height for larger aerosol concentration related to the inversion layer above cloud which prohibited the further growth of clouds?

   Answer: In the polluted conditions simulations the highest cloud tops penetrate the inversion layer, evaporate there and hence drive an increase in the inversion base height with time. However, as the initial inversion base height is similar in all simulations it cannot explain the differences in cloud top between the simulations at the initial stage. At aerosol concentration above 500 cm⁻³ the mean COG slightly decreases with aerosol loading since the negative effects of aerosol take over. Above this concertation the increase in evaporation efficiency at the cloud margin and the water loading take over and dominate the increase in the condensation efficiency at the cloud core and the decrease in effective terminal velocity. For the COG (as well as for the maximum cloud top height) the optimal aerosol concentration is higher than the one for the liquid water mass because it is more effected by the positive aerosol effects in the cloud core (Dagan et al., 2015a).

7. Line 259-260: The LWP is decreasing with larger aerosol concentration. Is the water loading larger?

Answer: We thank the reviewer for this comment. It is true that the mean LWP decreases in the simulations with aerosol concentration above 100 cm$^{-3}$. Below 100 cm$^{-3}$ an increase in the aerosol loading results in an increase in the mean LWP (Fig. 1B in the main text). For high aerosol loading (>100 cm$^{-3}$) both the LWP and the cloud fraction decrease with aerosol loading but for most vertical levels the liquid water content (LWC) increases.

Figure 4 in the text presents vertical profiles of the mean LWC for four simulations differ by the aerosol loading which implies about the water loading effect.

The increase in the water loading effect under polluted conditions is a result of the rain suppression (Fig. 1F) and the decrease in the effective terminal velocity (Fig. 3B) that drive the water mass to be pushed higher in the atmosphere.

Regarding the apparent contradiction between the trend in LWP (decreasing for aerosol concentration above 100 cm$^{-3}$ – Fig. 1B) and the mean LWC (generally increases with aerosol loading – Fig. 4). The values calculated in Fig. 4 are for the cloudy pixels. The LWC generally increases with height but the mean cloud fraction decreases (above the cloud base H~500m - see Fig. S2 below). Considering the two factors together (vertical profiles of CF and LWC) results in a decrease in LWP with aerosol concentrations above 100 cm$^{-3}$.

We added the figure below to the SI as well as explanations and clarifications regarding this point to the revised manuscript: *"We show that both the height and the magnitude of the maximum LWC increase with the aerosol loading. This is due to both rain suppression (Fig. 1F) and an increased $V_{COG}$ (Fig 3C) with aerosol loading. There is a reduction in the mean LWP (for >100 cm$^{-3}$ - Fig. 1B) although there is an increase in the LWC with aerosol loading due to the differences in cloud fraction (Fig. 1C, Fig. S2). Below the clouds' base (H<~550m) the LWC trend is reversed due to the enhancement of rain in the clean runs (Fig. 1F). The increase in LWC with aerosol loading implies a larger water loading negative component in the clouds' buoyancy."*

[Figure]

*Figure S2. Vertical profiles of the mean (over time and domain) cloud fraction (CF) per height for four simulations (with aerosol loading levels of 5 cm$^{-3}$ – blue, 50 cm$^{-3}$ – green, 250 cm$^{-3}$ – red, and 2000 cm$^{-3}$ – cyan).*

8. Line 297-299: Suggest to show 1-2 figures related the time variations of cloud fields to support the statements.

Answer: we thank the reviewer for the suggestion. Following this comment we have added to the new supporting information (SI) file a figure presenting the temporal evolution of the vertical profiles of the thermodynamic conditions (temperature and water vaper mixing ratio). Figure S3 (see below) presents the time evolution of the mean thermodynamic conditions for two simulations with aerosol loading of 50 cm$^{-3}$ (clean – upper row) and 2000 cm$^{-3}$ (polluted – lower row). This figure demonstrates that under clean conditions the cloudy layer (H~500-1500m) becomes warmer and dryer with time while the sub-cloud layer becomes colder and wetter. On the other hand, under polluted conditions the trend is different and the most significant change occurs at the inversion layer which significantly cools and becomes wetter with time.

These different changes in the thermodynamic conditions under clean and polluted conditions are the focus of another new paper (Dagan et al., 2016).

The addition to the revised manuscript: *"Figure 5 presents the changes (final value minus initial one) in the temperature (T) and water vapor content ($q_v$) vertical profiles as a function of aerosol concentration used in the simulation.*

*The initial profiles were identical in all simulations. Figure S3 (in the SI) presents the full temporal evolution of those parameters."*

The additions to the SI: "*Figure S3 presents the temporal change in vertical profiles of the temperature (left column) and water vapor mixing ratio (right column) for clean (50 $cm^{-3}$ – upper row) and polluted (2000 $cm^{-3}$ – lower row) conditions. It demonstrates that under clean conditions the cloudy layer (H~500-1500m) becomes warmer and dryer with time while the sub-cloud layer becomes colder and wetter. On the other hand, under polluted conditions the trend is different and the most significant change occurs at the inversion layer which significantly cools and becomes wetter with time".*

[Figure]

*Fig. S3 Temporal changes compared to the initial profiles of mean environmental temperature [K] (left) and mean water vapor mixing ratio [g/kg] (right). For two different simulations with aerosol concentrations of 50cm$^{-3}$ (clean – upper row) and 2000cm$^{-3}$ (polluted – lower row).*

9.  Line 331-335: Suggest to add more explanations to the results.

Answer: we thank the reviewer for this comment. Detailed explanations for this point are given at the next paragraph. We added clarification to the revised manuscript: "*Trends in the mean rain rate show that in the cleanest simulations (5, 25 and 50 $cm^{-3}$) it decreases with time (Fig. 1H, 53.3, 32.9 and 40.1%,*

*respectively). In the regime of medium to fairly high aerosol loading (100, 250 and 500 cm$^{-3}$) the rain rate increases (19.6, 598.1 and 841.5%, respectively). And in the most polluted simulations (2000 and 5000 cm$^{-3}$) the surface rain is negligible throughout the simulation time. These trends are explained below*

*The time evolution of the thermodynamic conditions (Fig. 5) shows a reduction (enhancement) in the thermodynamic instability with time in the clean (polluted) simulations. Figure 6 and table 1 indicate that under clean conditions the decrease in the thermodynamic instability with time leads to a decrease in the mean cloud field properties such as total mass, cloud top height and rain rate. Under polluted conditions the trends are opposite and the mean cloud field properties increase with time due to the increase in thermodynamic instability (Dagan et al., 2016) and due to the cloud deepening (Stevens and Seifert, 2008; Stevens, 2007; Seifert et al., 2015). These differences between the clean and polluted simulations drive changes in the optimum aerosol concentration with time. For example, for the LWP (Fig. 1B) the optimum aerosol concentration is 50, 100 and 250 cm$^{-3}$ for the first, second and third parts of the simulation, respectively."*

**Technical corrections:**

1. Line 35: add "water vapor and" at the end of this line;

   Answer: Following this comment, we have added the reviewer suggestion to the revised manuscript: "*Warm cloud (containing liquid water only) formation depends on the availability of water vapor and aerosols acting as cloud condensation nuclei (CCN).*"

2. Line 63: remove ","between the parentheses;

   Answer: Thank you for this comment. We have removed it: "*For aerosol concentrations lower than the optimum, the positive relationship between aerosol concentration and cloud development is a result of two main processes: 1) larger latent heat release driven by the increase in the condensation efficiency causing stronger updrafts, and 2) decrease in the*

*effective terminal velocity (η, i.e. mass weighted terminal velocity of the hydrometeors) (Koren et al., 2015) due to initial smaller droplets and the delay in the collision-coalescence process."*

3. Line 158: "in (Siebesma et al., 2003)"should be replaced by "by Siebesma et al. (2003)";

    Answer: We corrected it in the revised manuscript: *"This case was initialized using the setup specified in Siebesma et al. (2003)."*

4. Line 184-185: remove ();

    Answer: We have removed it in the revised manuscript: *"The LWP (Liquid Water Path - Fig. 1B) calculated as a mean over time over all cloudy columns in each domain, which is strongly correlated with the total water mass, also shows the same non-monotonic general trend."*

5. Line 295: change "less" to "minus";

    Answer: We corrected it in the revised manuscript*: "Figure 5 presents the changes (final value minus initial one) in the temperature (T) and water vapor content ($q_v$) vertical profiles as a function of aerosol concentration used in the simulation."*

6. Line 625: Add the variable for the abscissa.

    Answer: It was added in the revised manuscript:

[Figure]

*Figure 2. Domain mean condensation (solid lines) and evaporation (dashed lines) tendencies for four different simulations conducted with different aerosol concentration levels (5 cm$^{-3}$ blue, 50 cm$^{-3}$ green, 250 cm$^{-3}$ red and 2000 cm$^{-3}$ cyan).*

**Anonymous Referee #2**

The manuscript investigated cloud-aerosol interaction in warm cloud environment using a Large Eddy Simulations (LES) with bin microphysics. The paper is generally well written, contains some potentially interesting model results, but does not go into sufficient detail in any one of the cases and there is no comparison with observations which is necessary for a reality check. The article can be considered for publication after major revision.

Answer: we thank the reviewer for acknowledging our interesting results. We have changed the manuscript according to all of the reviewer's comments and we believe it presents the work in sufficient details now.

**Some specific comments and suggestions follow:**

1.  The algorithm of cloud-aerosol interaction in the model is not clear. Suggest to elaborate the section 2 (methodology) in details. How the aerosol initiations are taking place? What is the composition of aerosols they have considered in their model?

    Answer: We thank the reviewer for rising this important point. The model is initialized using an oceanic aerosol size distribution (Jaenicke, 1988; Altaratz et al., 2008). The concentrations we report are near the surface and the aerosols are assumed to maintain constant mixing ratio with height.

    The aerosols serve as potential cloud condensation nuclei (CCN) and activated based on Kohler theory (Khain et al., 2000). A prognostic equation for the aerosol is solved, which includes regeneration upon evaporation and removal by surface rain. The aerosols are assumed to be composed of ammonium-sulfate.

We added this information to the revised manuscript: *"The shape of the aerosol size distribution was based on measurements of marine aerosol distribution (see details in Jaenicke 1988 and Altaratz et al., 2008). Eight different simulations were conducted with a changing aerosol concentration (total concentration of: 5, 25, 50, 100, 250, 500, 2000 and 5000 cm$^{-3}$ near ground level, Dagan et al., 2015a). For reducing the sensitivity of our results to the aerosol size distribution shape and focusing on the aerosol number concentration effect, the different aerosol concentrations are calculated by multiplication of all bins in the smallest concentration size distribution by a constant factor and maintaining a similar shape of the size distribution. The aerosol is assumed to be composed of ammonium-sulfate and maintains constant mixing ratio with height. A prognostic equation is solved for the aerosol mass, including regeneration upon evaporation and removal by surface rain. Regeneration upon evaporation of cloud drops was shown to be a very important source of aerosols, especially in polluted conditions (Yin et al., 2005). The aerosol serves as potential cloud condensation nuclei (CCN) and it is activated based on the Kohler theory (Khain et al., 2000). The aerosol (water drop) size distribution is calculated between 5 nm to 2 μm (2 μm-3.2 mm). For both aerosol and drops, successive bins represent doubling of the mass."*

2. Does the model take nucleation scavenging? Need to be cleared.

   Answer: The prognostic equation for aerosol solved by the model does consider removal by surface rain. As was mentioned above, this information was added to the revised manuscript: *"A prognostic equation is solved for the aerosol mass, including regeneration upon evaporation and removal by surface rain."*

3. How does different size distribution of aerosols (CCN) affect the DSD? What is the effect of large CCN on the processes of droplet nucleation?

Answer: We thank the reviewer for this comment. For reducing the sensitivity of our results to the aerosol size distribution and focusing on the aerosol number concentration effect we have kept the shape of the aerosol size distribution constant in all different simulations. We have increased the aerosol number concertation by a constant factor, multiplying all bins. We added this information to the methodology section of the revised manuscript: "*For reducing the sensitivity of our results to the aerosol size distribution shape and focusing on the aerosol number concentration effect, the different aerosol concentrations are calculated by multiplication of all bins in the smallest concentration size distribution by a constant factor and maintaining a similar shape of the size distribution*".

In our case, as will be demonstrate later (answer no. 13), increasing the aerosol number concentration results in more and smaller cloud droplets. This can also be seen in Fig. 1 below which presents examples of aerosol and drop size distributions for four different simulations. The aerosol size distributions represent the initial conditions near the surface while the drop size distributions represent a level slightly above the cloud base (H=660m) after 2 hours of simulations. The drop size distribution is an average over all cloudy pixels (LWC>0.01 g/kg) with vertical velocity above 1 m/s.

Figure 1 below demonstrates that under polluted conditions there are more numerus and smaller droplets compared to clean conditions.

[Figure]

Fig. 1. Mean initial near surface aerosol size distribution (left column) and mean drop size distribution above cloud base (H=660m right column) after two hours of simulation for four different simulations (5cm$^{-3}$ – first row, 50cm$^{-3}$ – second row, 250cm$^{-3}$ – third row, and 2000cm$^{-3}$ – bottom row). The drop size distribution is an average over all cloudy pixels (LWC>0.01 g/kg) with vertical velocity above 1 m/s.

The sensitivity of the results presented here (especially the non-monotonic response of the cloud fields' mean properties) to different aerosol size distributions that do not include GCCN (giant CCN) is small. The largest aerosol radius bin used in this study is 2 µm and hence large GCCN are not considered here.

However, the response of rain amount to aerosol concentration is sensitive to the availability of GCCN (Dagan et al., 2015b; Feingold et al., 1999; Yin et al., 2000). The availability of GCCN can produce rain even under extremely polluted conditions. The existence of GCCN at the cloud base height can be considered as a special case. Khain et al. (2000) claimed that GCCN can affect the initiation of the collection process only if they reach the cloud base height in concentrations that are comparable to rain drop concentrations ($10^{-4}$-$10^{-3}$ cm$^{-3}$), which for large aerosol is not always trivial. Thus, for simplicity reasons, in this study we choose to avoid the GCCN effect. Moreover, even if GCCN are available, some reduction in rain amount is expected under high aerosol loading (Dagan et al., 2015b).

4. Authors have discussed on the role of aerosol on collision-coalescence, delayed surface rain in their introduction (line 53- 56) and also about cloud invigoration mechanism in line 80-83. In this context authors should put the references (Hazra et al. 2013a, Journal of Atmospheric Science; Hazra et al. 2013b, Journal of Geophysical Research-Atmosphere) studied cloud-aerosol interactions using 2 moment bulk microphysical scheme (Cheng et al. 2007, QJRMS) in meso-scale model. In this regard, I suggest refer/include all those papers in the manuscript.

Answer: we thank the reviewer for this comment. We added those references to the revised manuscript:

*"The initiation of collision-coalescence is delayed in polluted clouds (Gunn and Phillips, 1957; Squires, 1958; Albrecht, 1989). This drives a delay in rain formation and can affect the amount of surface rain (Rosenfeld, 1999, 2000; Cheng et al., 2007 ; Khain, 2009; Levin and Cotton, 2009; Koren et al., 2012; Hazra et al., 2013a;b; Dagan et al., 2015b).*

"*The invigoration mechanism, which refers to deeper and larger clouds with larger mass that develop under polluted conditions was studied mainly in deep convective clouds (Andreae et al., 2004; Koren et al., 2005; Rosenfeld et al., 2008; Tao et al., 2012; Fan et al., 2013; Hazra et al., 2013a; Altaratz et al., 2014). Our focus here is on warm cloud fields for which previous observational studies reported on invigoration effect or a non-monotonic response of the clouds to an increase in aerosol loading."*

5. Line 87-91: How aerosol impact on cloud fraction (CF), discussion not clear.

Answer: in this part of the introduction we discuss previous observational works that examined the relations between aerosol and warm convective clouds' properties. Cloud fraction (CF) was proposed to be affected by aerosol due to rain suppression and prolonging the cloud lifetime (Albrecht, 1989). In addition, warm convective cloud invigoration by aerosol (larger, deeper and stronger precipitating clouds under polluted conditions) also drive increase in the cloud fraction (Koren et al., 2014).

Following this comment we have added clarification to the revised manuscript: *"From convective stability considerations deeper clouds tend to have larger area (larger CF). It was shown that warm convective cloud's area correlates positively with cloud's depth (Benner and Curry, 1998; Koren et al., 2008)."*

6. Line: 169-172: Authors have mentioned the aerosols number concentration. Is it total concentration of all bins? Or which bin size of aerosols number concentrations they are increasing. This should be cleared to the readers.

    Answer: In this sentence we were referring to the total aerosol concentration. For simplicity reasons (as was explained in answer No. 3), the changes in aerosol concentration are done by multiplying all bins by a constant factor and maintaining a similar aerosol size distribution. We added clarification to the revised manuscript: *"Eight different simulations were conducted with a changing aerosol concentration (total concentration of: 5, 25, 50, 100, 250, 500, 2000 and 5000 $cm^{-3}$ near ground level, Dagan et al., 2015a). For reducing the sensitivity of our results to the aerosol size distribution shape and focusing on the aerosol number concentration effect, the different aerosol concentrations are calculated by multiplication of all bins in the smallest concentration size distribution by a constant factor and maintaining a similar shape of the size distribution."*

7. Line 177-179: Authors mentioned "the eight simulated cloud fields are examined first". Are the eight simulated cloud fields are in similar or different. Need to be explained.

    Answer: in this sentence we describe that at this section the aerosol effect on the mean cloud field properties are examined for eight simulations. The eight different simulations describe different aerosol concentrations. To make this point clearer, this sentence was revised: *"The aerosol effects on the mean properties of the eight simulated cloud fields (differ by the aerosol loading) are examined first."*

8. In Figure 1 and Figure 6: Total liquid water mass, LWP, CF and rain rate show a "tipping" point. It will be worthy if authors discuss on the "tipping point" as revealed in Figure 1 and 6. Another important point authors should explain why the "tipping point" for total water mass, LWP are different from the CF and rain rate.

Answer: We thank the reviewer for this comment. We agree with the reviewer that this "tipping point" is the heart of this paper and should be explained thoroughly.

We refer to this "tipping point" as a non-monotonic response with an optimal aerosol concentration and try to put this result in the focus of this paper (starting from the title). The non-monotonic trends with an optimal aerosol concentration of the mean cloud field properties are explained by contradicting aerosol effects. Namely, two positive aerosol effects: 1) increase in the condensation efficiency (as shown in Fig. 2 of the main text), and 2) increase in the droplet mobility (as shown in Fig. 3 in the main text). And two negative aerosol effects: 1) increase in the evaporation efficiency (as shown in Fig. 2 of the main text), and 2) increase in the water loading (as shown in Fig. 4 in the main text). All of those effects are discussed and presented in the paper.

Regarding the different optimal aerosol concentration for the different cloud field properties, in a previous study (Dagan et al., 2015a) it was shown that more core oriented cloud properties have a higher optimal aerosol concentration than periphery based properties. For example, the strongest vertical velocities (W) in the domain are more frequent at the clouds' cores in which the conditions can be considered as closer to adiabatic. As a result W will be more affected by the increase in the condensation efficiency with aerosol loading (more related to the cloud core) then by the increase in the evaporation efficiency (more related to the cloud periphery) and will have high optimal aerosol concentration (Fig. 3A). A different example is the total cloud mass which is determine by both the cloud core and cloud periphery and hence affected by both the positive and negative aerosol effects. As a result the total cloud mass will have lower optimal aerosol concentration. Each of the different cloud field properties presented in figures 1 and 6 is a result of a different blend of core and periphery oriented processes and hence has a different optimal aerosol concentration.

This is being explained in the text: "*The minor decrease seen for this range of aerosol concentration (compared with the larger decrease in the mean LWP for example) can be explained by the location of the maximal cloud top height above the cloud core, which is affected mainly by the invigoration processes (enhanced condensation and latent heat release) and less by margin oriented processes (enhanced entrainment and evaporation) that significantly impact the total cloud mass (Dagan et al., 2015a). Another reason is the cloud deepening effect under polluted conditions (Stevens, 2007; Seifert et al., 2015) that will be described later.*"

9. Can authors consider another aerosol concentration at lower rage?

Answer: We thank the reviewer for this comment. The lowest aerosol concentration used in this study is 5 $cm^{-3}$ which is already extremely low and is not common in the natural environment. We have used an extremely large range of aerosol concentrations (3 orders of magnitudes, between 5 and 5000 $cm^{-3}$). Which is much larger then in many of the recent cloud-aerosol interactions studies (e.g. Seifert et al., 2015 who used a range of 35-105 $cm^{-3}$ – factor of three). In addition, we have used relatively high resolution (8 different aerosol concentration) especially at the lower range (5, 25, 50, 100, 250, 500, 2000 and 5000 $cm^{-3}$).

Under aerosol concentration lower than 5 $cm^{-3}$ we expect a very limited clouds' development due to the very low aerosol concentration. The total droplet surface area will be extremely low and hence the condensational growth of the drops will be very non-efficient. In addition, the low amount of droplets would result in insignificant competition between the droplets on the available water vapor and hence they would grow fast and start to precipitate early. The early initialization of precipitation will act as a positive feedback and reduce even more the droplet surface area and condensation rate.

10. Authors have showed the response of vertical wind (updraft) to varying CCN concentrations (Figure 3A). It shows updraft is monotonically increasing with aerosol concentrations. Is it correct? Need to be explained.

Answer: Increasing aerosol loading result in an increase in the condensation efficiency at the cloud core (super-saturated volumes) and hence increase in the latent heat release. This results in an increase in the buoyancy and vertical velocity in the polluted clouds. In Fig. 3A the mean weighted by the mass vertical velocity is presented for being consistent with the COG point of view. This mass-weighted mean represent more the cloud core then the cloud margin (which has less mass). As was explained in answer 8, the strongest vertical velocities (W) in the domain are more frequent at the clouds' cores in which the conditions can be considered as closer to adiabatic. As a result W will be more effected by the increase in the condensation efficiency with aerosol loading (more related to the cloud core) then by the increase in the evaporation efficiency (more related to the cloud periphery) and will have high optimal aerosol concentration. At high aerosol concentrations (above 2000 cm$^{-3}$) this effect saturates and the mean updraft remains similar.

This is being explained in the text: *"The mean updraft (in both space and time, weighted by the liquid water mass in each grid box to be consistent with the COG point of view - Fig. 3A) increases with the increase in aerosol loading, in agreement with previous studies (Saleeby et al., 2015; Seigel, 2014). This indicates an increase in the latent heat contribution to the cloud buoyancy, driven by increase in the condensation efficiency (Dagan et al., 2015a,b; Koren et al., 2014; Pinsky et al., 2013; Seiki and Nakajima, 2014) (Fig. 2)."*

11. Another doubt: Figure 1B and Figure 4: Figure 1B shows clear tipping point of LWP, first it increases then it decreases, whereas when I am looking into the Figure 4, LWC increasing with aerosol number concentrations. Authors should make it unambiguous.

Answer: The apparent contradiction between the trend in LWP (decreasing for aerosol concentration above 100 cm$^{-3}$ – Fig. 1B) and the mean LWC (generally increase with aerosol loading – Fig. 4) is because of the different cloud fraction in the different heights (see Fig. S2 below). The values calculated in Fig. 4 are for the cloudy pixels. The LWC generally increase with height but the mean cloud fraction decreases (above the cloud base H~500m - Fig. S2). Considering the two factors together (vertical profiles of

CF and LWC) results in a decrease in LWP with aerosol loading at concentrations above 100 cm$^{-3}$.

The increase in the water loading effect under polluted conditions is a result of the rain suppression (Fig. 1F) and the decrease in the effective terminal velocity (Fig. 3B) that drive the water mass to be pushed higher in the atmosphere.

We added the figure below to the SI as well as explanations and clarifications regarding this point to the revised manuscript: *"We show that both the height and the magnitude of the maximum LWC increase with the aerosol loading. This is due to both rain suppression (Fig. 1F) and an increased $V_{COG}$ (Fig. 3C) with aerosol loading. There is a reduction in the mean LWP (for >100 cm$^{-3}$ - Fig. 1B) although there is an increase in the LWC with aerosol loading due to the differences in cloud fraction (Fig. 1C, Fig. S2). Below the clouds' base (H<~550m) the LWC trend is reversed due to the enhancement of rain in the clean runs (Fig. 1F). The increase in LWC with aerosol loading implies a larger water loading negative component in the clouds' buoyancy."*

[Figure]

*Figure S2. Vertical profiles of the mean (over time and domain) cloud fraction (CF) per height for four simulations (with aerosol loading levels of 5 cm$^{-3}$ – blue, 50 cm$^{-3}$ – green, 250 cm$^{-3}$ – red, and 2000 cm$^{-3}$ – cyan).*

12. Time dependent part (Figure 6, line 338-349) is not much convincing. Need to be explained in details.

Answer: we thank the reviewer for this comment. Following this comment we have added additional information to the revised manuscript to improve clarity. Figure S3 (see below) presents the temporal evolution of the vertical profiles of the thermodynamic conditions (temperature and water vaper mixing ratio) for two simulations with aerosol loading of 50 cm$^{-3}$ (clean – upper row) and 2000 cm$^{-3}$ (polluted – lower row). This figure demonstrates that under clean conditions the cloudy layer (H~500-1500m) becomes warmer and dryer with time while the sub-cloud layer becomes colder and wetter. Those changes drive reduction of the thermodynamic instability with time. On the other hand, under polluted conditions the trend is different and the most significant change occurs at the inversion layer which significantly cools and becomes wetter with time. Those changes drive increase in the thermodynamic instability with time.

The differences in the thermodynamic evolution under clean and polluted conditions drive changes in the mean cloud field properties (Fig. 6). Under clean conditions the thermodynamic instability decreases and hence the mean cloud field properties (total mass, CF, mean LWP) also decrease with time. On the other hand, under polluted conditions the thermodynamic instability increases with time and hence also the mean cloud field properties.

These different changes in the thermodynamic conditions under clean and polluted conditions are the focus of another new paper (Dagan et al., 2016).

The addition to the revised manuscript: *"Figure 5 presents the changes (final value minus initial one) in the temperature (T) and water vapor content (q$_v$) vertical profiles as a function of aerosol concentration used in the simulation. The initial profiles were identical in all simulations. Figure S3 (in the SI) presents the full temporal evolution of those parameters."*

The additions to the SI: *"Figure S3 presents the temporal change in the temperature (left column) and water vapor mixing ratio (right column) vertical profiles for clean (50 cm$^{-3}$ – upper row) and polluted (2000 cm$^{-3}$ – lower row) conditions. It demonstrates that under clean conditions the cloudy layer (H~500-1500m) becomes warmer and dryer with time while the sub-*

*cloud layer becomes colder and wetter. On the other hand, under polluted conditions the trend is different and the most significant change occurs at the inversion layer which significantly cools and becomes wetter with time".*

[Figure]

*Fig. S3. Temporal changes compared to the initial profiles of mean environmental temperature [K] (left) and mean water vapor mixing ratio [g/kg] (right). For two different simulations with aerosol concentrations of 50cm$^{-3}$ (clean – upper row) and 2000cm$^{-3}$ (polluted – lower row).*

13. It will be worthy if authors provide cloud drop number concentration and cloud drop size with increasing aerosol concentration.

    Answer: we thank the reviewer for this comment. Following this comment we have added the manuscript supporting information file (SI) some information about the cloud drop mean size and concentration for the different simulations (differ by the aerosol loading). Information about the aerosol and drop size distribution is presented in answer number 3 above. Figure S1 below presents the vertical profiles of the mean cloud drop concentration and size. It demonstrates that at the cloudy layer (H>500m) the mean drop size decreases with the aerosol loading, while its concentration increases (Twomey, 1977). Below cloud base the trend is reversed – larger concentration and smaller rain drops under clean conditions (Altaratz et al., 2008).

The additions to the revised manuscript: *"The effects of changes in aerosol concentration on the drop concentration and its mean size, for the different simulations can be found in Fig. S1 in the supporting information (SI)."*

The additions to the SI: *"Figure S1 presents vertical profiles of the mean concentration and mean cloud drop size per height. It demonstrates that at the cloudy layer (H>500m) the mean drop size decreases with aerosol loading, while its concentration increases (Twomey, 1977). Below cloud base the trend is reversed – larger rain drops and lower concentration under more polluted conditions (similar to what was shown in Altaratz et al., 2008)."*

[Figure]

*Figure S1. Vertical profiles of a) the mean (over time and domain) drop radius, and b) the mean (over time) of drop maximum concentration. These results include both cloud and raindrops for four simulations (with aerosol loading levels of 5 cm$^{-3}$ – blue, 50 cm$^{-3}$ – green, 250 cm$^{-3}$ – red, and 2000 cm$^{-3}$ – cyan).*

14. Can authors put some hints of cloud-aerosol interactions in the mixed-phase clouds in their "Conclusion" section?

    Answer: We thank the reviewer for this comment. The results presented in this study are for the aerosol effect of the mean properties of warm convective cloud fields. Warm processes act as the initial and boundary conditions for mixed-phase processes in deep convective clouds. Hence, understanding the

aerosol effect on the warm processes is the first step in understanding aerosol effects on deep convective clouds. We can expect that the aerosol effect on the warm processes (condensation-evaporation, collection etc.) in deep convective clouds would be similar to the ones described here for warm convective clouds (especially during the first stages of the cloud, when there are only warm processes). However, ice processes are much more complicated and much is still unknown about them so taking them into consideration making the aerosol effect much harder for prediction. The additional phase transition and the interactions between phases add many levels of complexity. Great part of the challenges in understanding aerosol effects on deep convective clouds is attributed to the fact that ice nucleation is far from being fully understood (DeMott et al., 2015). Ice nucleation processes are shown to be extremely sensitive to the aerosol surface properties in a way that is not yet understood (Vali, 2014) and many freezing schemes are based on empirical relations that are far from being converged to one comprehensive theory. Therefore we have chosen to gain first a detailed process level understanding of the warm phase.

Nevertheless, we expect the general trend to be similar. Regarding the thermodynamic feedback for example, in deep convective clouds, higher aerosol loading was shown to result in larger water mass being pushed to higher levels (i.e. Hazra et al., 2013a; Koren et al., 2010; Rosenfeld et al., 2008; Storer and van den Heever, 2012, among many others). This is expected to result in more evaporation at higher levels, hence cooling and moistening of those levels and increasing the atmospheric instability. Since the convective clouds that pass the freezing level are usually thicker, we expect the optimal aerosol concentrations to be higher (Dagan et al., 2015a). And indeed, aerosols were reported to increase rain amount from deep convective clouds (Hazra et al., 2013a; Koren et al., 2012; Li et al., 2008). This could lead to the opposite result of warming of the cloudy layer and cooling of the sub-cloud layer by rain evaporation, hence consumption of the atmospheric instability. Determining the end result of those two contradicting effects cannot be done a-priory and should be investigated in details. Because this subject is highly speculative and the fact that we do not show anything regarding aerosol effect on deep convective clouds in this manuscript, we prefer not to include any

speculations about it in the discussion and to keep this subject to future studies.

15. Line 35-36: It is not only depends on CCN concentrations, also depends on availability of water vapor.

    Answer: We thank the reviewer for this comment. We changed this sentence in the revised manuscript: "*Warm cloud (containing liquid water only) formation depends on the availability of water vapor and aerosols acting as cloud condensation nuclei (CCN).*"

16. The authors probably should proof read the manuscript carefully to correct some typos

    Answer: we thank the reviewer for this comment. The manuscript has been gone thought proof reading.

**References**

Albrecht, B. A. (1989), Aerosols, cloud microphysics, and fractional cloudiness, *Science (New York, NY)*, *245*(4923), 1227.

Altaratz, O., I. Koren, T. Reisin, A. Kostinski, G. Feingold, Z. Levin, and Y. Yin (2008), Aerosols' influence on the interplay between condensation, evaporation and rain in warm cumulus cloud, *Atmospheric Chemistry and Physics*, *8*(1), 15-24.

Benner, T. C., and J. A. Curry (1998), Characteristics of small tropical cumulus clouds and their impact on the environment, J. Geophys. Res., 103(D22), 28753–28767, doi:10.1029/98JD02579.;

Berg, W., T. L'Ecuyer, and S. van den Heever (2008), Evidence for the impact of aerosols on the onset and microphysical properties of rainfall from a combination of satellite observations and cloud-resolving model simulations, *Journal of Geophysical Research: Atmospheres (1984–2012)*, *113*(D14).

Cheng, C.-T., W.-C. Wang, and J.-P. Chen (2007), A modelling study of aerosol impacts on cloud microphysics and radiative properties, *Quarterly Journal of the Royal Meteorological Society*, *133*(623), 283-297.

Dagan, G., I. Koren, and O. Altaratz (2015), Competition between core and periphery-based processes in warm convective clouds–from invigoration to suppression, *Atmospheric Chemistry and Physics*, *15*(5), 2749-2760.

Dagan, G., I. Koren, and O. Altaratz (2015), Aerosol effects on the timing of warm rain processes, *Geophysical Research Letters*, *42*(11), 4590-4598.

Dagan, G., Koren, I., Altaratz, O., and Heiblum, R. H.: Aerosol effect on the evolution of the thermodynamic properties of warm convective cloud fields, Scientific Reports, in press, 2016.

DeMott, P. J., A. J. Prenni, G. R. McMeeking, R. C. Sullivan, M. D. Petters, Y. Tobo, M. Niemand, O. Möhler, J. R. Snider, and Z. Wang (2015), Integrating laboratory and field data to quantify the immersion freezing ice nucleation activity of mineral dust particles, *Atmospheric Chemistry and Physics*, *15*(1), 393-409.

Feingold, G., W. R. Cotton, S. M. Kreidenweis, and J. T. Davis (1999), The impact of giant cloud condensation nuclei on drizzle formation in stratocumulus: Implications for cloud radiative properties, *Journal of the Atmospheric Sciences*, *56*(24), 4100-4117.

Hazra, A., B. Goswami, and J.-P. Chen (2013a), Role of interactions between aerosol radiative effect, dynamics, and cloud microphysics on transitions of monsoon intraseasonal oscillations, *Journal of the Atmospheric Sciences*, *70*(7), 2073-2087.

Hazra, A., P. Mukhopadhyay, S. Taraphdar, J. P. Chen, and W. R. Cotton (2013b), Impact of aerosols on tropical cyclones: An investigation using convection-permitting model simulation, *Journal of Geophysical Research: Atmospheres*, *118*(13), 7157-7168.

Jaenicke, R.: Aerosol physics and chemistry, Landolt-Börnstein Neue Serie 4b, 391–457, 1988.

Khain, A. P., M. Ovchinnikov, M. Pinsky, A. Pokrovsky, and H. Krugliak (2000), Notes on the state-of-the-art numerical modeling of cloud microphysics, Atmos. Res., 55(3–4), 159–224, doi:10.1016/S0169-8095(00)00064-8.

Koren, I., Oreopoulos, L., Feingold, G., Remer, L. A., and Altaratz, O.: How small is a small cloud?,Atmos. Chem. Phys., 8, 3855-3864, 2008

Koren, I., G. Dagan, and O. Altaratz (2014), From aerosol-limited to invigoration of warm convective clouds, *science*, *344*(6188), 1143-1146.

Seifert, A., T. Heus, R. Pincus, and B. Stevens (2015), Large-eddy simulation of the transient and near-equilibrium behavior of precipitating shallow convection, *Journal of Advances in Modeling Earth Systems*.

Seigel, R. B. (2014), Shallow Cumulus Mixing and Subcloud Layer Responses to Variations in Aerosol Loading, *Journal of the Atmospheric Sciences*(2014).

Small, J. D., P. Y. Chuang, G. Feingold, and H. Jiang (2009), Can aerosol decrease cloud lifetime?, *Geophysical Research Letters*, *36*(16).

Stevens, B.: On the growth of layers of nonprecipitating cumulus convection, Journal of the atmospheric sciences, 64, 2916-2931, 2007.

Twomey, S. (1974), Pollution and the planetary albedo, *Atmospheric Environment (1967)*, *8*(12), 1251-1256.

Twomey, S. (1977), The influence of pollution on the shortwave albedo of clouds, *Journal of the atmospheric sciences*, *34*(7), 1149-1152.

Vali, G. (2014), Interpretation of freezing nucleation experiments: singular and stochastic; sites and surfaces, *Atmospheric Chemistry and Physics*, *14*(11), 5271-5294.

Xue, H. W., and G. Feingold (2006), Large-eddy simulations of trade wind cumuli: Investigation of aerosol indirect effects, *Journal of the Atmospheric Sciences*, *63*(6), 1605-1622.

Xue, H. W., G. Feingold, and B. Stevens (2008), Aerosol effects on clouds, precipitation, and the organization of shallow cumulus convection, *Journal of the Atmospheric Sciences*, *65*(2), 392-406.

Yin, Y., K. S. Carslaw, and G. Feingold. "Vertical transport and processing of aerosols in a mixed-phase convective cloud and the feedback on cloud development." *Quarterly Journal of the Royal Meteorological Society* 131.605 (2005): 221-245.

Yin, Y., Z. Levin, T. G. Reisin, and S. Tzivion (2000), The effects of giant cloud condensation nuclei on the development of precipitation in convective clouds—a numerical study, *Atmospheric research*, *53*(1), 91-116.

---

## Author Response (AR2)

**Replay to the review of: "Time dependent, non-monotonic response of warm convective cloud fields to changes in aerosol loading"**

Dear Patrick,

Thank you so much for taking the time and reviewing our paper. It is not trivial that editor takes the lead and invests the time to carefully review a paper. We appreciate it. Your comments indeed helped us improving the clarity of the paper. We believe that after the additional clarifications and the modifications, the presented results are much clearer.

Specifically you highlighted the results presented in Figure 2 of the paper. As you'll see in the detailed reply section below, there is no mistake in the figure. There are two reasons for the differences between the numbers we are showing to the ones in Lee et al, (2012). In Lee et al, you show a domain average for all voxels. Here we wanted to show the aerosol effect on the cloudy voxels only. We show that aerosols amplify both the evaporation and condensation processes in the polluted runs. Sometimes, there is a cancelation of the overall effect (condensation-evaporation) and so the net numbers are much smaller but the effect that the diffusion processes are enhanced means that the dynamic can be enhanced as well and so many other feedbacks. In our paper we averaged for cloudy voxels only, separately for supersaturation (condensation) and subsaturation (evaporation). As a sanity check we averaged as in Lee et al, and got similar results. Following this comment we have changed the figure and added explanations to make our results clearer.

Below please find detailed point-by-point answers to all of your comments. Please note that in the answers we present 3 types of figures. Figures that appear in the main text (numbered as Fig. x), figures that appear in the supporting information file (SI - numbered as Fig. Sx) and figures that appear only in this reply (numbered as Fig. Rx).

Lines 46-49: How do you define condensation or evaporation efficiency? Is it a column-integrated value? Later on (line 262) you reference Pinsky et al. 2013 and Seiki and Nakajima 2014, but in neither paper does the word "efficiency" actually appear. Do these papers use a different term? Are these really appropriate references if the term "condensation efficiency" does not appear? Koren et al. 2014 mentions

efficiency once but does not define the term. Please define this term carefully at some point, as you use it in various parts of the manuscript.

**Answer:** Thank you for highlighting this point. The condensation efficiency is related to the characteristic time scale of condensation (or evaporation) discussed in the provided references. As shown in Fig. 3 in Pinsky et al. (2013) and in Fig. 1 in Seiki and Nakajima (2014) the characteristic time scale of condensation (or evaporation) depends negatively on the cloud droplet number concentration. It means that under polluted conditions for a given supersaturation (subsaturation) the condensation (evaporation) will be faster. In other words, under polluted conditions the supersaturation will be consumed faster and more efficiently, increasing the latent heat release and invigorating the cloud development. To make it clearer we added a clarification to the revised manuscript:

*"Under given supersaturation conditions, the condensation in polluted clouds is more efficient (higher condensation rate or shorter consumption time of the supersaturation - Pinsky et al., 2013; Seiki and Nakajima, 2014; Koren et al., 2014; Kogan and Martin, 1994; Dagan et al., 2015a)."*

154-157: This doesn't very well describe how this manuscript is different from the numerous previous studies. Please make this clear.

**Answer:** Following this comment we have changed this part in the revised manuscript: *"In this work we explore the coupled microphysical-dynamic system of warm marine cloud fields using a bin-microphysics scheme under a large range of aerosol concentrations. We study the aerosol-cloud-environmental thermodynamic system by examining how changes in aerosol concentrations affect clouds properties, the related modifications of the thermodynamic conditions over time which as well drive feedbacks on the clouds' properties evolution."*

170-171: Only 2 hrs for spin-up? Lee et al. 2012 (in their Fig. 1; note: I was a co-author on this paper) shows 4 to 6 hrs was necessary for their case. Are you sure these

simulations are properly spun-up? Also, do the simulations include a diurnal cycle? Or is it fixed sun zenith angle? Or is it night time?

**Answer:** In order to better understand the complex interplay between microphysics, cloud dynamics and the derived evolution of the environmental thermodynamic conditions we preferred to simplify the system and to focus on idealized cases without including interactive radiative effects. Radiative cooling and changes of the SST due to changes in short wave radiation would be very important for marine stratocumulus cases for which the cloud fraction is large and therefore the radiative component is a key player. However, here since the average cloud fraction is low (~15%) and in order to separate coupled competing processes, the aerosol effects on the evolution of the thermodynamic properties were considered only by their effects via clouds. The radiative effects are prescribed and included in the large scale forcing (LSF - see table R1 below). To make this point clearer we added clarification to the methodology part of the revised manuscript: *"In order to focus on the aerosol effect on the thermodynamic properties of the field, the radiative effects (as included in the large scale forcing see details in Dagan et al., 2016) were prescribed in all simulations"*.

As for the spin-up time, the figures below present the time evolution of the total liquid water mass in the domain (Fig. R1a) and the domain mean liquid water center of gravity height (COG, Fig. R1b) for four different simulations differ by aerosol concentrations. After less than 2 hours of simulation the initial increase in these values desists and the difference between the simulations becomes significant. We note that other BOMEX studies (e.g. (Xue and Feingold, 2006;Jiang et al., 2006;Grabowski, 2014), and other trade Cu case studies (Xue et al., 2008;Seifert et al., 2015) used 2 h as the spin-up time. Following this comment we added a clarification to the revised manuscript: *"After 2 h of simulations the initial increase in the total liquid water mass in the domain desisted and the differences between the simulations (differ by the aerosol loading) became significant. Therefore 2h is determined as spin-up time (similar to the spin-up time in Xue and Feingold, 2006)."*

[Figure]

Figure R1. Evolution of (a) Total liquid water mass and (b) liquid water center of gravity for four different simulations differ by the aerosol loading.

Table R1. The large-scale forcing (LSF) standard setup used for the BOMEX case study simulations. The LSF includes temperature, water vapor mixing ratio and vertical velocity tendencies as a function of height.

| Z [m] | Temperature tendency by LSF [K/s] | Water vapor mixing ratio tendency by LSF [kg/kg s] | Vertical velocity tendency by LSF [m/s] |
|---|---|---|---|
| 0 | $-2.315 \cdot 10^{-5}$ | $-1.2 \cdot 10^{-8}$ | 0 |
| 300 | $-2.315 \cdot 10^{-5}$ | $-1.2 \cdot 10^{-8}$ | -0.0013 |
| 500 | $-2.315 \cdot 10^{-5}$ | 0 | -0.0021 |
| 1500 | $-2.315 \cdot 10^{-5}$ | 0 | -0.065 |
| 2100 | $-1.389 \cdot 10^{-5}$ | 0 | 0 |
| 2500 | 0 | 0 | 0 |

175: So what is the fixed shaped of the aerosol size distribution? "based on..." is not particularly enlightening.

**Answer:** The aerosol size distribution used in this work is a marine aerosol size distribution, which was published in Jaenicke 1988 and was described in Altaratz et al., 2008, as well. We give details in the text about the method we used to change the concentration but keep the size distribution similar. To make it clearer we changed this sentence in the revised manuscript: *"The aerosol distribution adopts a marine size distribution (see details in Jaenicke 1988 and Altaratz et al., 2008). Eight different simulations were conducted simulating a wide range of aerosol loading conditions from extremely pristine to polluted (total concentration of: 5, 25, 50, 100, 250, 500, 2000 and 5000 cm$^{-3}$ near ground level, Dagan et al., 2015a). To reduce the results sensitivity to the shape of the aerosol size distribution and to focus on the aerosol number concentration effect, the different aerosol concentrations are calculated by multiplication of all bins by a constant factor and maintaining a similar shape of the size distribution."*

178: Some of these values are really very small. Are they really applicable to the atmosphere? (Also see comment regarding line 200 about adding Nd to Fig 1).

**Answer:** Thank you for this comment. On the same note of our answer regarding the radiative effects, we think that using ideal conditions helps gaining better process level understanding and factor separation. Some of the assumptions are not common in nature (concentration of 5 cm$^{-3}$ is indeed super pristine). However, learning how the system will react to the extreme clean (or polluted) conditions helps constraining competition between processes. Moreover, some of the systems sensitivities to aerosol concentration are of logarithmic nature in which changes from super pristine to slightly polluted can yield the clearest aerosol signal, which can later be explored for the whole aerosol spectrum.

We added to the revised text: *"Eight different simulations were conducted simulating a wide range of aerosol loading conditions from extremely pristine to polluted (total*

*concentration of: 5, 25, 50, 100, 250, 500, 2000 and 5000 cm$^{-3}$ near ground level, Dagan et al., 2015a)."*

184-185: I'm confused by the fact that the aerosol "maintains constant mixing ratio with height" but simultaneously "a prognostic eqn is solved for the aerosol mass". If aerosol mass can appear because of, say, cloud drop evaporation, how can aerosol mass mixing ratio be constant with height? If aerosol size distributions are held constant, this is of course not physically realistic... what are the consequences for interpreting these results in the context of the real atmosphere?

**Answer:** Indeed our description was not clear enough. Only on the initialization stage of the simulation the aerosol mixing ratio is setup as constant with height. Once the simulation starts running the prognostic aerosol equation determines the aerosol mixing ratio at each voxel. To make this point clearer we added clarification to the revised manuscript: *"The aerosol is assumed to be composed of ammonium-sulfate and initialized with constant mixing ratio with height. A prognostic equation is solved for the aerosol mass, including regeneration upon evaporation and removal by surface rain."*

189: Is Khain et al. 2000 a reference to Kohler theory, or a scheme to calculate activation? If the former, that seems like a poor choice.

**Answer:** the aerosol activation scheme we used is described in Khain et al. 2000 and is based on the Kohler theory. In the revised text we added: *"The aerosol serves as potential cloud condensation nuclei (CCN) and it is activated based on the Kohler theory (the scheme is described in Khain et al., 2000)."*

200: Analyzing the data in this way supposes that there's no overall tendency in time. Is this true? If not, what is the tendency relative to the de-trended variability? How would the results change if a different time window (say, 8 hrs instead of 14) were analyzed?

**Answer:** This question is answered on the next section of the paper. As first order view (in section 3.1) we calculate the mean cloud field properties for the entire

simulation time. In the following section (3.2) we divide the simulations to shorter time windows and examine the trends with time (please see Fig. 6 in the manuscript). We added a clarification sentence about it: *"The aerosol effects on the mean field properties during the entire run are examined first following by a more detailed examination of the time evolution in the next section".*

200: Please add a panel showing mean drop concentration.

**Answer:** based on the previous reviewers' comments we added information regarding the mean drop concentration and size to the supporting information (SI). Since our paper is already loaded with text and figures, we preferred to leave it in the SI.

The reference to it in the main text: *"The effects of changes in aerosol concentration on the drop concentration and its mean size, for the different simulations can be found in Fig. S1 in the supporting information (SI)."*
The information provided in the SI:

***"S.1 Mean size and number of drops***
*Figure S1 presents vertical profiles of the mean concentration and mean drop size per height. It demonstrates that at the cloudy layer (H>500m) the mean drop size decreases with aerosol loading, while its concentration increases (Twomey, 1977). Below cloud base the trend is reversed – larger rain drops and lower concentration under more polluted conditions (similar to what was shown in Altaratz et al., 2008).*

[Figure]

[Figure]

*Figure S1. Vertical profiles of a) the mean (over time and domain) drop radius, and b) the mean (over the second two hours of simulation - after the spin-up time) of drop maximum concentration. These results include both cloud and raindrops for four simulations (with aerosol loading levels of 5 cm$^{-3}$ – blue, 50 cm$^{-3}$ – green, 250 cm$^{-3}$ – red, and 2000 cm$^{-3}$ – cyan). Doted parts of the curves represent heights in which the total liquid water mass was less than 1% of the maximum total mass (Fig. 4b) to avoid conclusions based on small statistics."*

200: What is the number of clouds in any given simulation? Does it change with aerosol number? Perhaps add this as a separate panel to Fig. 1.

**Answer:** recent studies showed that the number of clouds in the domain increases with the aerosol loading (Heiblum et al., 2016;Seigel, 2014). The small amount of clouds under clean conditions is a result of clustering of the clouds under precipitating conditions (Seifert and Heus, 2013). This trend is also true in our simulations (see (Heiblum et al., 2016) which is based on the same set of simulations). Nevertheless, it was also shown that in order to fully capture the organization pattern one must simulates a larger domain (Seifert and Heus, 2013) which is beyond our current computational abilities (using bin microphysical scheme). This limitation is mentioned in the main text: *"Due to computational limitations, we had to restrict the*

*domain size to a scale that has a limited capability for capturing large scale organization (Seifert and Heus, 2013)."*

Because of this limitation and because it was the focus of a few recent studies (some of which are mentioned above) we prefer not to focus on warm cumulus organization and its reflection on the total cloud number in this study.

218: Choosing the maximum value of any parameter seems very prone to outliers. Could you instead plot the, say, 90th or 95th percentile cloud top height to tell the same story?

**Answer:** We agree that maximum values can be sometimes prone to outliers. In our case however, we present the mean over time of the maximum cloud top height. This averaging is based on 1 minute time series over 14 hours (840 values) and hence possible contribution of outliers is averaged out. Moreover, we present in Figures 1 and 6 both the maximum and the mean cloud top height. We believe that both of them together capture, in the best way, the cloud fields' mean properties response to changes in aerosol loading (together with the total cloud mass, LWC and CF).

Moreover, the maximum cloud top height is a predictor of the cloud deepening effect over time (which is discussed in section 3.2). The size distribution of the clouds contains many small clouds and only few large ones, which contribute to the evaporation at the inversion layer and the subsequent deepening of the cloudy layer. Those clouds are really at the tail of the distribution and hence we prefer to keep the maximum cloud top.

To show that the $90^{th}$ percentile cloud top height demonstrates the same general trend (both for the mean over the entire simulation time and for the evolution in time as presented in Fig. 6 of the main text) we present it here as well (Fig R2).

We have added clarification to the revised manuscript: *"Presenting together the mean over time of the maximum and the mean cloud top height captures, in a compact, yet informative, way the response of the clouds top height distribution in the domain to changes in aerosol loading and reduces the sensitivity to outliers. Moreover, by averaging over time we decreased the significance of the outliers as well."*

[Figure]

Figure R2. The 90th percentile cloud top height (CTH) as a function of the aerosol concentration used in the simulation. In addition to the total (black curve) it is calculated separately for each period of one third of the simulations (blue, green and red for the first, second and third periods, respectively). The error bars present the standard error.

242: I don't understand Fig 2. Lee et al. 2012 shows values between NET condensation of clean and polluted simulations to be on the order of 0.1 to 1 g/kg/day, which is about 1e-6 to 1e-5 g/kg/s. You show (gross) values that are as much as 4 orders of magnitude larger. These are not compatible. I think something is very wrong with your plot.

**Answer:** Thank you for this comment that helped us clarifying the paper's findings. There is no contradiction between the two results. The reason for the differences between Lee et al. (2012) and our results is the averaging method. As in many cases there is a choice whether to average for the entire domain that includes mostly voids (in the case of trade Cu) or to average for cloudy pixels only. Moreover, one can present the total condensation-evaporation balance, or since both are measurable and take place in other voxels in the cloud, one can present both condensation and evaporation rates, separately. Lee et al. (2012) presented the domain's mean net condensation evaporation while we present separately the mean gross values for the rate of condensation and evaporation for cloudy voxels only. We used this method of averaging to emphasis the increase in both condensation (in the supersaturated areas in the domain) and evaporation (in the subsaturated areas in the domain) rates with

aerosol loading, despite the decrease in cloud fraction (as can be seen in Fig. 1C in the main text and Fig. R4 below). Figure R3 below presents the mean values (over space and time) of net condensation rate for our simulations (using the same averaging method as preformed in Lee et al. 2012) and indeed its order of magnitude is 1e-5 g/kg/s. The small differences between Lee et al. (2012) and our results can be probably attributed to the different case study simulated (RICO vs BOMEX) and to the different microphysical scheme.

[Figure]

Figure R3: Domain's mean net condensation-less-evaporation tendencies for four different aerosol loading levels (5 cm$^{-3}$ – blue, 50 cm$^{-3}$ – green, 250 cm$^{-3}$ – red, and 2000 cm$^{-3}$ – cyan).

Following this comment and to avoid confusion we have moved the original Fig. 2 into the SI (as it supports our claim of an increase in both condensation and evaporation rates with aerosol loading) and now we present an alternative figure in the main text (please see below) that shows the vertical profiles of the total condensation and evaporation mass for the different simulations.

The changes done accordingly in the revised manuscript: "*Fig. 2 presents the vertical profiles of the total condensation and evaporation mass during the simulations, for four different simulations. We note that as the aerosol loading increases, both the*

*condensed and evaporated mass increased (this is due to the increase in the diffusion rates – see Fig. S2, SI, and despite the decrease in cloud fraction – see Fig. 1C, Dagan et al., 2015a; Koren et al., 2014; Pinsky et al., 2013; Seiki and Nakajima, 2014). Below cloud base (located around 550 m) the clean simulations have small rain evaporation values which is absent in the polluted simulations."*

[Figure]

Figure 2 (the revised version). Domain's total condensed (solid lines) and evaporated mass (dashed lines) for 14 hours of simulation along four different simulations conducted with different aerosol concentration levels (5 cm$^{-3}$ blue, 50 cm$^{-3}$ green, 250 cm$^{-3}$ red and 2000 cm$^{-3}$ cyan).

The additions to the SI:

*"**S.2 Mean condensation and evaporation rates***

*Figure S2 presents vertical profiles of the mean (over time) of the condensation and evaporation rates, per height, for four simulations with different aerosol loading. It demonstrates the increase in both condensation and evaporation rates with aerosol loading.*

[Figure]

*Figure S2. Domain's mean condensation (solid lines) and evaporation (dashed lines) tendencies for four different simulations conducted with different aerosol concentration levels (5 cm$^{-3}$ blue, 50 cm$^{-3}$ green, 250 cm$^{-3}$ red and 2000 cm$^{-3}$ cyan). Note that doted parts of the curves represent heights in which the total liquid water mass was less than 1% of the maximum total mass (Fig. 4b)."*

Second, why is this value so much larger at high altitude than at low altitude, esp for the 2000 cm$^{-3}$ case. Cloud base is somewhere around 500 m (Fig 4), but the mean condensation rate is something like 2 orders of magnitude higher at altitude (2000 to 3000 m). For an adiabatic parcel, the condensation rate generally goes DOWN (albeit not very quickly) with altitude because of the exponential dependence of the saturation mixing ratio on T with the possible exception of the tens of meters near cloud base where there's substantial supersaturation. Also, the difference between the cond and evap tendencies is huge in this area for the 2000 cm$^{-3}$ case. This suggests that LWC should increase greatly at these altitudes (since precip is absent). But Fig. 4 shows that LWC is roughly constant with height in this simulation. I don't think these two are compatible.

**Answer:** As mentioned above the values calculated in the previous version of Fig. 2 are for the cloudy voxels only. In this case the values of the condensation and evaporation rates increase with height because of the changes in cloud fraction as a function of height (see Fig. R4 below). Naturally, the condensation values were

calculated only for the supersaturated fraction of the domain while the evaporation values are only for the subsaturated fraction.

At the higher end of the vertical cloud extension (H>2000m) the cloud fraction is extremely low (<1%) and the subset of supersaturated voxels is smaller. Only the parcels with the strongest vertical velocity and which are closest to adiabatic conditions (high LWC) still have supersaturation at those heights. In those extreme cases the condensation rate can be relatively high. The mean condensation rate for this small fraction of the domain (even if it has large values) is translated into small total condensed mass when looking on the entire domain and integrating over the time (see the new Fig. 2 above). Moreover, the subsaturated fraction of the domain at these heights is larger than the supersaturated fraction and therefore the net condensation-evaporation rate in the polluted simulations is negative.

Following these comments, we added to Fig. 4 in the main text a panel that presents the vertical profile of the total liquid water mass in the domain (Fig. 4b, see below). Figure 4a in the main text presents the mean LWC only for the cloudy voxels while Fig. 4b presents the total water mass per height. Examining both of them together demonstrates that despite the increase in the mean water loading in the upper parts of the clouds with aerosol loading, the total water mass decreases at those layers. Moreover, in order to highlight that the higher parts of the profiles contain small fraction of cloudy voxels, we have marked the heights in which the total mass is less then 1% of the maximum mass (based on Fig. 4b). This was done both in Fig. 4a in the main text and in Figs S1 and S2 in the SI.

[Figure]

Figure R4. Vertical profiles of the mean (over time and domain) cloud fraction (CF) per height for four simulations (with aerosol loading levels of 5 cm$^{-3}$ – blue, 50 cm$^{-3}$ – green, 250 cm$^{-3}$ – red, and 2000 cm$^{-3}$ – cyan).

The new Fig. 4 in the main text:

[Figure]

*Figure 4. a) Mean liquid water content (LWC) vertical profiles. b) Vertical profiles of the mean (over time) total liquid water mass per height for four different simulations (5 cm$^{-3}$ blue, 50 cm$^{-3}$ green, 250 cm$^{-3}$ red and 2000 cm$^{-3}$ cyan). The mean profiles are calculated for the last 14 hours out of the 16 hours of simulation. Note that doted parts of the carves in a) represents heights in which the total liquid water mass was less then 1% of the maximum total mass (Fig 4b).*

To prevent this confusion, we have changed Fig. 2 to present the total condensed and evaporated mass (and not the mean rates per cloudy voxels). Now, some of the signal is reduced due to the reduction in cloud fraction with aerosol loading.

Lastly, the 1 to 2 orders of magnitude difference between aerosol cases seems wrong. There can be differences (see again Lee et al. 2012) but I don't think they can be that big. By my reasoning, something is very wrong with this plot.

**Answer:** This bring us back to the answer related to what is presented in the plots. As explained above we do not want to show only the net effect but instead part of this paper's message is that aerosols increase the rates in both directions. Increasing the aerosol concentration increases both condensation and evaporation. Sometimes the net effect can be almost balanced (or be relatively small) not sowing this enhancement that in our opinion is responsible to an important part of the aerosol effect. When examining the aerosol effect on the net condensation and evaporation tendencies for the entire domain and the entire time it includes competing affects that reduce the total signal. Specifically, increasing aerosol loading increases both the condensation and evaporation rates and decreases the mean cloud fraction. As explained above, following the editor's comments we have decided to present vertical profiles of the total condensed and evaporated mass which shows a factor of 2-3 difference between clean and polluted simulations.

To avoid the same kind of confusion between averages taking on the whole domain vs. those on cloudy voxels, we also added clarification to the revised manuscript regarding Fig. 4:

*"We show that both the height and the magnitude of the maximum LWC increase with the aerosol loading. This is due to both rain suppression (Fig. 1F) and an increased $V_{COG}$ (Fig. 3C) with aerosol loading. There is a reduction in the mean LWP (for >100 $cm^{-3}$ - Fig. 1B) although there is an increase in the LWC with aerosol loading due to the differences in cloud fraction (Fig. 1C) and in the vertical distribution of the liquid water (Fig. 4b). At the upper part of the clouds (H>2000m), in the polluted case, the small amount of cloudy pixels have a large mean LWC (and hence a large water*

*loading effect) but the total amount of liquid water is small (Fig. 4b). Below the clouds' base (H<~550m) the LWC trend is reversed due to the enhancement of rain in the clean runs (Fig. 1F). The increase in LWC with aerosol loading implies a larger water loading negative component in the clouds' buoyancy."*

Fig 1: Panel B does not have units of kg/m2. Please include some "uncertainty bars" to indicate the time variability in all panels.

**Answer:** we thank the editor for this comment. The units in panel B were changed to $g/m^2$. The error bars in Fig. 1 present the standard errors which are much smaller than the mean values and than the differnces between the simulations.

[Figure]

**Figure 1. mean properties (over domain and time) of the simulated cloud fields as a function of the aerosol concentration used in the simulation: A) total liquid water mass in the domain, B) cloudy LWP, C) cloud fraction (CF) for columns with $\tau>0.3$, D) maximum cloud top, E) mean cloud top, and, F) surface rain rate. Each of these mean properties are calculated for the last 14 hours out of the 16 hours of simulation. The error bars present the standard errors. For details about the different properties see the text.**

[Figure]

**Figure 6. Mean properties (over time and domain) of the simulated cloud fields as a function of the aerosol concentration used in the simulation: A) total liquid water mass in the domain, B) cloudy LWP, C) cloud fraction (CF) for columns with τ>0.3, D) maximum cloud top, E) mean cloud top, and, F) surface rain rate. Each property is calculated separately for each period of one third of the simulations (blue, green and red for the first, second and third periods, respectivly). The error bars present the standard error. For details about the different properties, see the text.**

**Thanks again for the important comments. We hope that the revised manuscript is clearer.**

**References**

Grabowski, W. W.: Extracting Microphysical Impacts in Large-Eddy Simulations of Shallow Convection, Journal of the Atmospheric Sciences, 71, 4493-4499, 2014.

[revised manuscript text omitted]

---

## Author Response (AR3)

**Replay to the review of: "Time dependent, non-monotonic response of warm convective cloud fields to changes in aerosol loading"**

Dear Dr. Ervens,

We would like first to thank you for agreeing to take our paper and to complete the review process so fast, it is highly appreciated. We also appreciate the time and efforts you have put in reading the revised manuscript and our previous responses to the referee comments. Please find below a point by point answers to your comments.

l. 10: Change 'properties' to 'loading' as you do not explore effects of any other aerosol properties (composition, size)

**Answer:** Thank you for this correction. It was changed: *"Large Eddy Simulations (LES) with bin microphysics are used here to study cloud fields' sensitivity to changes in aerosol loading and the time evolution of this response."*

2. 191: If the aerosol size distribution is only scaled up/down, the shape should be identical, not similar

**Answer:** Thank you. Indeed the aerosol size distribution is constant. We have corrected it in the revised manuscript: *"To reduce the results sensitivity to the shape of the aerosol size distribution and to focus on the aerosol number concentration effect, the different aerosol concentrations are calculated by multiplication of all bins by a constant factor and maintaining a constant shape of the size distribution."*

3. 341: How does the study by Dagan et al., 2016, differ from the current one?
**Answer:** In Dagan et al. (2016) we did not discuss the aerosol effect on the mean cloud field properties and their non-monotonic trend but only the thermodynamic evolution. Changes in the thermodynamic conditions do change the cloud scale processes and specifically the transition from cloud enhancement to suppression (i.e. the evolution of the non-monotonic trend). Therefore this current paper is dedicated to show the interplay between the evolution of the cloud field thermodynamic properties and their interactions with the cloud scale non-monotonic behavior. Specifically, in this study we examine the response of cloud fields' mean properties to changes in the aerosol loading. This is done both globally (during the entire simulation period, Sec. 3.1) and for different periods along the simulation (Sec. 3.2). We show that the mean field properties change in a non-monotonic trend, with an optimal aerosol concentration that can be explained by contradicting aerosol effects on processes that encourage cloud development versus those that suppress it. The time evolution of this response and the increase in time of the optimal aerosol concentration are driven by the evolution of the thermodynamic conditions that is different for different aerosol loading conditions.

In line 341 we mentioned that the focus of Dagan et al. (2016) is the changes in the thermodynamic evolution under different aerosol concentrations: "*All the aerosol effects that were discussed up to this point (condensation-evaporation efficiencies, η and water loading) are applicable both on the single cloud scale as well as on the cloud field scale. However, on the cloud field scale, another aspect needs to be considered, namely the time evolution of the effect of clouds on the field's thermodynamic conditions (which was the focus of a recent study by Dagan et al., 2016).*"

4. 374: either 'last' or 'third' seems redundant here

**Answer:** We have changed it in the revised manuscript: *"On the other hand, in the more polluted simulations, (with aerosol loading of 250 and 500 cm$^{-3}$) there is an increase in the total water mass with time (of 17 and 37% between the first and the last periods of the simulations, respectively)."*

5. 381: a) There is no Figure 1F

b) I am confused (but this might be due to the missing figure): The rain rate is given in mm/day (e.g. fig. 1F). How does this translate into percentages?

**Answer:** Thank you. Indeed this was a mistake. We corrected it in line 381 to "Fig. 6F". The revised manuscript: *"Trends in the mean rain rate show that in the cleanest simulations (5, 25 and 50 cm$^{-3}$) it decreases with time (Fig. 6F, 53.3, 32.9 and 40.1%, respectively). In the regime of medium to fairly high aerosol loading (100, 250 and*

*500 cm$^{-3}$) the rain rate increases (19.6, 598.1 and 841.5%, respectively). And in the*

*most polluted simulations (2000 and 5000 cm$^{-3}$) the surface rain is negligible*

*throughout the simulation time. These trends are explained below."*

As for all properties presented in Fig. 6 and table 1 we calculated the percentile change between the last and first part of the simulation for better understanding its time evolution. It is explained in the text: "*
[revised manuscript text omitted]